# Deep Reinforcement Learning with Stacked Hierarchical Attention for Text-based Games

**Yunqiu Xu**[*]
University of Technology Sydney
Yunqiu.Xu@student.uts.edu.au

**Meng Fang**[*]
Tencent Robotics X
mfang@tencent.com

**Ling Chen**
University of Technology Sydney
Ling.Chen@uts.edu.au

**Yali Du**
University College London
yali.du@ucl.ac.uk

**Joey Tianyi Zhou**
IHPC A*STAR
zhouty@ihpc.a-star.edu.sg

**Chengqi Zhang**
University of Technology Sydney
Chengqi.Zhang@uts.edu.au

## Abstract

We study reinforcement learning (RL) for text-based games, which are interactive simulations in the context of natural language. While different methods have been developed to represent the environment information and language actions, existing RL agents are not empowered with any reasoning capabilities to deal with textual games. In this work, we aim to conduct explicit reasoning with knowledge graphs for decision making, so that the actions of an agent are generated and supported by an interpretable inference procedure. We propose a stacked hierarchical attention mechanism to construct an explicit representation of the reasoning process by exploiting the structure of the knowledge graph. We extensively evaluate our method on a number of man-made benchmark games, and the experimental results demonstrate that our method performs better than existing text-based agents.

## 1 Introduction

Language plays a core role in human intelligence and cognition [14, 43]. Text-based games [13, 20], where both the states and actions are described by textual descriptions, are suitable simulation environments for studying the language-informed decision making process. These games can be regarded as an intersection of natural language processing (NLP) and reinforcement learning (RL) tasks [35]. To solve text-based games via RL, the agent has to tackle many challenges, such as learning representation from text [42], making decisions based on partial observations [4], handling combinatorial action space [57] and sparse rewards [56]. Generally, existing agents for text-based games can be classified as rule-based agents and learning-based agents. Rule-based agents, such as NAIL [21], solve the games based on pre-defined rules, engineering tricks, and pre-trained language models. By heavily relying on prior knowledge of the games, these agents lack flexibility and adaptability. With the progress of deep reinforcement learning [38, 39], learning-based agents such as LSTM-DRQN [42] become increasingly popular since they learn purely from interaction without requiring expensive human knowledge as prior. Recently, considering the rich information that can be maintained by its structural memory, knowledge graphs (KGs) have been incorporated into RL agents to facilitate solving text-based games [1, 4, 3].

---

[*]Equal contribution.

While a lot of studies have been conducted on representing useful information from text observations [3, 4, 42] and reducing action spaces [20, 57], few RL agent addresses the reasoning process for text-based games. Going beyond mapping a question to an answer, human beings have the ability of reasoning − they can reuse the knowledge [50], or compose the supporting facts (e.g., the relation between objects in the scene) from the question and the knowledge base to interpret the answer [10, 30]. We believe that RL agents empowered with reasoning capabilities will be better mimicking human decisions in solving text-based games and achieving enhanced performance. In terms of RL agents, we consider enhancing the reasoning capability of the agent by exploiting KGs. While existing studies [3, 4, 58] treat KGs as a part of the observation to handle partial observability, they ignore the potential of KGs for reasoning [12, 27]. Furthermore, the effectiveness of reasoning is constrained by two problems. Firstly, existing KG-based agents construct one single KG, so that fine-grained information (e.g., the types of object relationship, the newness/oldness of information) is hard to be maintained. Secondly, the multi-modal inputs, such as textual observations and KGs, are aggregated via simple concatenation so that their respective benefits cannot be sufficiently exploited.

We believe that an intelligent agent should have the ability to conduct explicit reasoning with relational and temporal awareness being taken into consideration to make decisions. In this paper, our goal is to design an enhanced RL agent with a reasoning process for text-based games. We propose a new method, named as **S**tacked **H**ierarchical **A**ttention with **K**nowledge **G**raphs (SHA-KG)[2], to enable the agent to perform multi-step reasoning via a hierarchical architecture on playing games. Briefly, to leverage the structure information of a KG that maintains the agent's knowledge about the game environment, we first consider the sub-graphs of the KG with different semantic meanings so that relational and temporal awareness will be taken into account. Secondly, a stacked hierarchical attention module is devised to build effective state representation from multi-modal inputs, so that their respective importance will be considered.

Our contributions include four aspects. Firstly, our work is a first step in pursuing reasoning in solving text-based games. Secondly, we propose to incorporate sub-graphs of the KG into decision making to introduce the reasoning process. Thirdly, we propose a new stacked hierarchical attention mechanism for RL approach featured by multi-level and multi-modal reasoning. Fourthly, we extensively evaluate our method on a wide range of text-based benchmark games, achieving favorable results compared with the state-of-the-art methods.

## 2 Related work

**Agents for text-based games.** Existing agents either perform based on predefined rules or learn to make responses by interacting with the environment. Rule-based agents [8, 16, 21, 31] attempt to solve text-based games by injecting heuristics. They are thus not flexible since a huge amount of prior knowledge is required to design rules [20]. Learning-based agents [2, 20, 22, 26, 42, 55, 56, 57] usually employ deep reinforcement learning algorithms to deliver adaptive game solving strategies. However, the performance of these agents is still not up to par when playing complex man-made games, even though efforts have been made to reduce the difficulty (e.g., DRRN [22] assumed that an action can only be selected from a valid action set for each state). KG-based agents have been developed to enhance the performance of learning-based agents with the assistance of KGs. KGs can be constructed by simple rules so that it substantially reduces the amount of prior knowledge required by rule-based agents. While KGs have been leveraged to handle partial observability [3, 4, 58], reduce action space [3, 4], and improve generalizability [1, 5], few of the existing works addresses its potential for reasoning. Recently, Murugesan et al. [41] tried to introduce commonsense reasoning for playing synthetic games. They extracted sub-graphs from ConceptNet [44], which is a large-scale external knowledge base with millions of edges and nodes. In contrast, we aim to construct the KG based on domain information with minimal external knowledge. Besides, we focus on man-made games which are more complex than synthetic games in terms of logic, so that the reasoning ability becomes especially crucial and desirable to the agents.

**Attention mechanism.** Attention mechanism has been widely studied in areas of machine learning, psychology and neuroscience [33]. For text-based games, self-attention [45] has been applied to encode textual observation [1, 58], and Graph Attention Networks (GATs) [46] has been employed to encode KGs [3]. Regarding model explainability, the attention mechanism helps to solve the outcome

explanation problem, e.g. building an attention-based saliency map [19]. For RL, the attention mechanism has been used to interpret the decision making process, mostly in tasks with visual inputs [18, 40]. For tasks with multi-modal inputs, such as visual question answering (VQA) [7], the attention mechanism has been used to aggregate the image and text inputs [23, 24, 29, 30, 34, 36]. In this work, we apply an attention mechanism to consider the multi-modal inputs based on text observations and graph structures.

**Reasoning via knowledge graph.** A lot of existing studies have used KGs as the knowledge base to facilitate learning and interpretation, including incorporating KG-based commonsense reasoning for question answering [9, 15, 32, 61] and recommendation [47, 48, 52, 60]. We are the first to exploit KGs to induce reasoning for RL-based agents playing text-based games.

## 3 Preliminaries

**POMDP** Partially Observable Markov Decision Processes (POMDPs) can be defined as a 7-tuple: the state set $\mathcal{S}$, the action set $\mathcal{A}$, the state transition probabilities $\boldsymbol{T}$, the reward function $\boldsymbol{R}$, the observation set $\boldsymbol{\Omega}$, the conditional observation probabilities $\boldsymbol{O}$ and the discount factor $\boldsymbol{\gamma} \in (0, 1]$. At each time step, the agent will receive an observation $\boldsymbol{o}_t \in \boldsymbol{\Omega}$, depending on the current state and previous action via the conditional observation probability $O(\boldsymbol{o}_t|\boldsymbol{s}_t, \boldsymbol{a}_{t-1})$. By executing an action $\boldsymbol{a}_t \in \mathcal{A}$, the environment will transit into a new state based on the state transition probability $T(\boldsymbol{s}_{t+1}|\boldsymbol{s}_t, \boldsymbol{a}_t)$, and the agent will receive the reward $\boldsymbol{r}_{t+1} = R(\boldsymbol{s}_t, \boldsymbol{a}_t)$. Same as Markov Decision Process (MDPs), the goal of the agent is to learn an optimal policy $\boldsymbol{\pi}^*$ to maximize the expected future discounted sum of rewards from each time step: $\boldsymbol{R}_t = \mathbb{E}[\sum_{k=0}^{\infty} \boldsymbol{\gamma}^k \boldsymbol{r}_{t+k+1}]$.

**KG** Knowledge Graph (KG) for a text-based game can be built from a set of triplets $\langle Subject, Relation, Object \rangle$, denoting that the $Subject$ has $Relation$ with the $Object$. For example, $\langle Kitchen, Has, Food \rangle$. The KG is denoted as $G = (V, E)$, where $V$ and $E$ are the node set and the edge set, respectively. Both $Subject$ and $Object$ belong to the node set $V$. $Relation$, which corresponds to the edge connecting them, belongs to $E$.

## 4 Methodology

### 4.1 Problem statement

In this work, we focus on man-made games, which are initially designed for human players [20]. These games are devised with more complex logic and much larger action space than synthetic games [13]. Text-based games require an agent to make automatic responses to achieve specific goals (e.g., escaping from the dungeon) based on received textual information. Raw textual observation contains only the feedback of taking an action (e.g., "Taken" is a textual observation after executing the action "take egg"). As underlying states can not be directly observed by the agent, the text-based games can be formulated as POMDPs. Similar to [3], at every step we construct an input $\boldsymbol{s}_t$ as the combination of three components: a textual observation $\boldsymbol{o}_{t,\text{text}}$, a collected raw score $\boldsymbol{o}_{t,\text{score}}$, and a KG $\boldsymbol{o}_{t,\text{KG}}$ (note that here $\boldsymbol{s}_t$ should not be regarded as a true game state as the games are not fully observable). $\boldsymbol{o}_{t,\text{text}}$ further includes the current state $\boldsymbol{o}_{t,\text{desc}}$ (describing the environment), inventory $\boldsymbol{o}_{t,\text{inv}}$ (describing items collected by a player), game feedback $\boldsymbol{o}_{t,\text{feed}}$, and previous action taken $\boldsymbol{a}_{t-1}$. Fig. 1 (a) shows an example of $\boldsymbol{o}_{t,\text{text}}$.

While $\boldsymbol{o}_{t,\text{text}}$ and $\boldsymbol{o}_{t,\text{score}}$ mainly reflect the current observation, $\boldsymbol{o}_{t,\text{KG}}$ records the game history. Therefore, the KG can help the agent to handle partial observability. At each time step, the triples extracted from the current textual observation $\boldsymbol{o}_{t,\text{text}}$ are used to update the KG as,

$$\boldsymbol{o}_{t,\text{KG}} = \text{GraphUpdate}(\boldsymbol{o}_{t-1,\text{KG}}, \boldsymbol{o}_{t,\text{text}}) \tag{1}$$

Fig. 1 (b) shows an example of $\boldsymbol{o}_{t,\text{KG}}$ and how it updates. We provide details of constructing and updating the KG in Sec. 5.2.

### 4.2 Sub-graph division

As discussed, existing KG-based agents build only one knowledge graph [1, 3, 4, 58]. To introduce relational-awareness and temporal-awareness, in this work, we divide our KG as multiple sub-graphs.

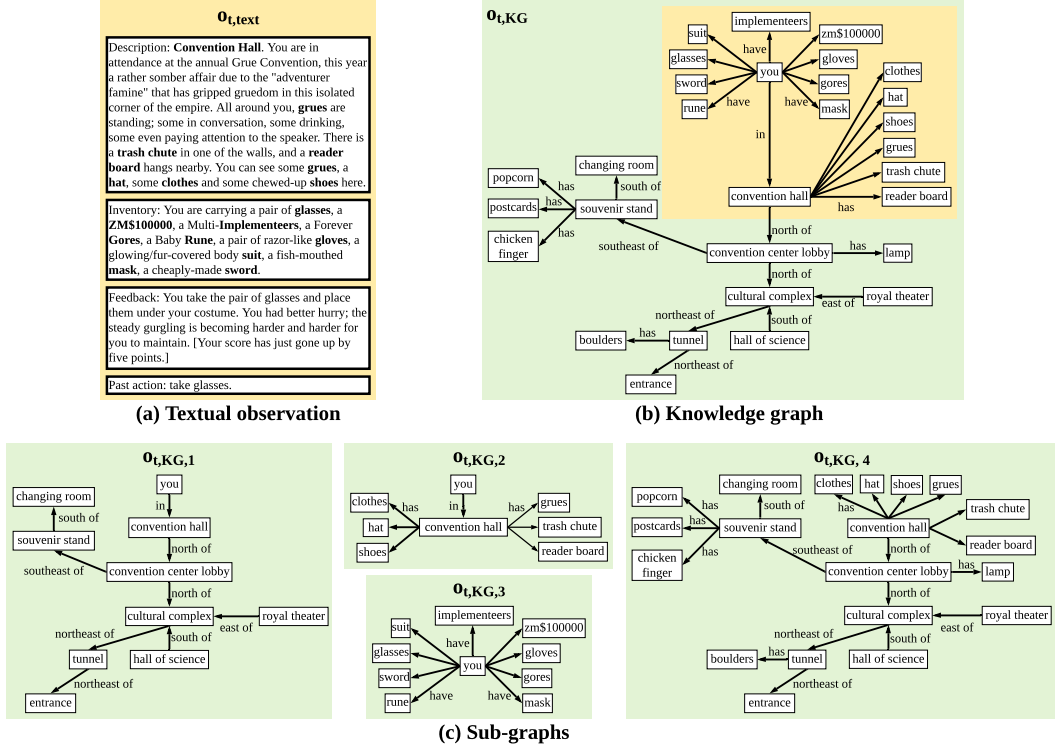

**(a) Textual observation**  **(b) Knowledge graph**

**(c) Sub-graphs**

Figure 1: (a) Textual observation $o_{t,\text{text}}$. (b) Knowledge graph $o_{t,\text{KG}}$. Yellow region in $o_{t,\text{KG}}$ contains information extracted from the observation $o_{t,\text{text}}$. (c) Sub-graphs obtained from $o_{t,\text{KG}}$.

Inspired by the heterogeneous graph [49, 59] where a graph contains different types of nodes and edges, we first classify edges by their types (e.g., "*Has*" and "*East of*" can be regarded as different types), and then build relational-aware sub-graphs based on the edges. In addition, as the KG can not distinguish between the current and past information, we introduce temporal-awareness via building different sub-graphs based on whether the historical information is included (e.g., sub-graphs built from $o_{t,\text{text}}$ only and sub-graphs built from $o_{t,\text{text}}$ and $o_{t-1,\text{KG}}$). The union of all sub-graphs is the full KG:

$$o_{t,\text{KG}} = o_{t,\text{KG},1} \cup o_{t,\text{KG},2} \ ... \ \cup o_{t,\text{KG,m-1}} \cup o_{t,\text{KG,m}} \tag{2}$$

where $m$ denotes the number of sub-graphs. From the perspective of hierarchical learning [54, 62], the sub-graph division allows observations to be considered in two levels: In the high level, the full KG captures the overall node connectivity. In the low level, the sub-graphs reflect different relational and temporal relations. Fig. 1 (c) shows an example of the sub-graphs obtained from $o_{t,\text{KG}}$.

### 4.3 Stacked hierarchical attention network

Before action selection, we represent the input $s_t$ as a vector $v_t$ first. We omit the subscript "t" for simplicity, and denote the KG as $o_{\text{KG,full}}$ to distinguish it from the sub-graphs. Since the textual observation, score and knowledge graph are multi-modal inputs, inspired by the VQA techniques [29, 34], we aggregate the inputs by constructing query representation for one modal (or two) to obtain the attention of another modal, through a stacked hierarchical attention mechanism. Fig. 2 shows an overview of our encoder, which consists of two levels. In the high level, we build a query vector from the KG and score, then compute multiple groups of attention values across the components of textual observations. In the low level, we treat the output of the high level as a query, and compute attention values across the sub-graphs.

**High-level attention** Similar to KG-A2C [3], the KG $o_{\text{KG,full}}$ is processed via GATs [46] followed by a linear layer to get the graph representation $v_{\text{KG, full}} \in \mathbb{R}^{d_{\text{KG}}}$. The score representation $v_{\text{score}} \in \mathbb{R}^{d_{\text{score}}}$ is obtained via binary encoding. While previous works [3, 20] concatenated all observational vectors to form state representation, we build the query vector $q_{\text{high}} \in \mathbb{R}^{d_{\text{high}}}$ by concatenating $v_{\text{KG, full}}$ and

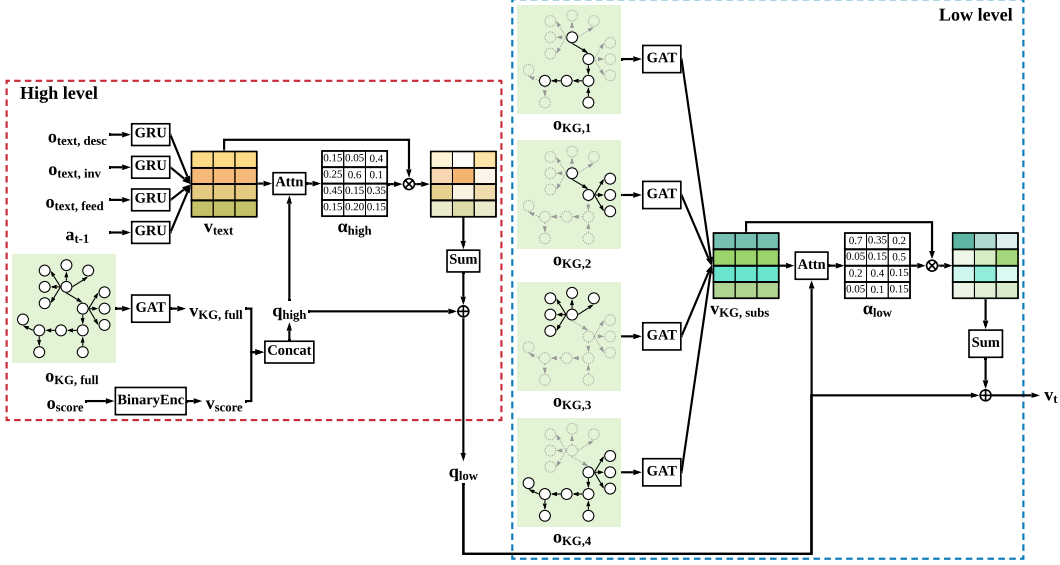

Figure 2: Overview of our stacked hierarchical attention network. In the high level (left), the query vector $q_{\text{high}}$ is the combination of the KG representation $v_{\text{KG, full}}$ and the score representation $v_{\text{score}}$. Then multiple groups of attention values $\alpha_{\text{high}}$ are computed across the components of textual observation $v_{\text{text}}$. In the low level (right), the query vector $q_{\text{low}}$ is the output of high level encoding, and multiple groups of attention values $\alpha_{\text{low}}$ are computed across the sub-graphs. The final output $v_t$ is served as the state representation for action selection.

$v_{\text{score}}$ followed by a linear layer:

$$q_{\text{high}} = W_{\text{Init}}\text{concat}(v_{\text{KG, full}}, v_{\text{score}}) + b_{\text{Init}} \tag{3}$$

where $W_{\text{Init}} \in \mathbb{R}^{d_{\text{high}} \times (d_{\text{KG}}+d_{\text{score}})}$ and $b_{\text{Init}} \in \mathbb{R}^{d_{\text{high}}}$ are weights and biases.

Suppose the textual observation consists of $c$ components[3], we first encode them separately by $c$ GRUs. Instead of concatenating, we consider them individually to build the textual representation vector $v_{\text{text}} \in \mathbb{R}^{d_{\text{high}} \times c}$. Therefore $v_{\text{text}}$ can be treated as multiple image regions or image representation with multiple channels. Inspired by SCA-CNN [11], we compute attention values in channel-wise. However, instead of computing one attention value for each channel, we compute multiple groups of attention values to capture more fine-grained information. Specifically, one group of attention values is computed for each position along the channel:

$$\alpha_{\text{high}} = \text{softmax}(W_{\text{A,high}}h_{\text{high}} + b_{\text{A,high}}) \tag{4}$$

where

$$h_{\text{high}} = \tanh(W_{\text{I,high}}v_{\text{text}} \oplus (W_{\text{Q,high}}q_{\text{high}} + b_{\text{Q,high}})) \tag{5}$$

denotes the intermediate representation and $\oplus$ denotes the addition of a matrix and a vector. $W_{\text{I,high}} \in \mathbb{R}^{d_{\text{high}} \times d_{\text{high}}}$, $W_{\text{Q,high}} \in \mathbb{R}^{d_{\text{high}} \times d_{\text{high}}}$ and $W_{\text{A,high}} \in \mathbb{R}^{d_{\text{high}} \times d_{\text{high}}}$ are weight matrices, $b_{\text{Q,high}} \in \mathbb{R}^{d_{\text{high}}}$ and $b_{\text{A,high}} \in \mathbb{R}^{d_{\text{high}}}$ are biases. This operation is equivalent to dividing $v_{\text{text}}$ as $d_{\text{high}}$ sub-vectors $v_{\text{text,sub}} \in \mathbb{R}^{1 \times c}$, then computing channel-wise attention for each of them. The obtained attention values $\alpha_{\text{high}} \in \mathbb{R}^{d_{\text{high}} \times c}$ reflect the multi-positional attentive focus on the textual components. The final step of high-level encoding is to attentively aggregate the query vector with the textual vector. In order to enable multi-level reasoning, we leverage recent advances in attention techniques [17, 28, 53] to learn multi-step reasoning by iteratively updating the query. We first multiply $v_{\text{text}}$ with $\alpha_{\text{high}}$ via dot-product, then sum all the channels and add it to $q_{\text{high}}$ to obtain updated query vector $q_{\text{low}} \in \mathbb{R}^{d_{\text{high}}}$:

$$q_{\text{low}} = q_{\text{high}} + \sum_{i}^{c} \alpha_{\text{high},i} \odot v_{\text{text},i} \tag{6}$$

**Low-level attention** The low-level encoding process is similar to high level, except that the attention values are computed across different sub-graphs. We encode sub-graphs with different GATs and combine them as graph representation $\boldsymbol{v}_{\mathrm{KG}} \in \mathbb{R}^{d_{\mathrm{low}} \times m}$, where $d_{\mathrm{low}}$ denotes dimensionality. We treat the output vector of high-level computing as a query vector, and perform linear transformation to ensure $\boldsymbol{q}_{\mathrm{low}} \in \mathbb{R}^{d_{\mathrm{low}}}$. Then we apply the similar attention mechanism of high-level:

$$\boldsymbol{\alpha}_{\mathrm{low}} = \mathrm{softmax}(\boldsymbol{W}_{\mathrm{A,low}}\boldsymbol{h}_{\mathrm{low}} + \boldsymbol{b}_{\mathrm{A,low}}) \tag{7}$$

where

$$\boldsymbol{h}_{\mathrm{low}} = \tanh(\boldsymbol{W}_{\mathrm{I,low}}\boldsymbol{v}_{\mathrm{KG}} \oplus (\boldsymbol{W}_{\mathrm{Q,low}}\boldsymbol{q}_{\mathrm{low}} + \boldsymbol{b}_{\mathrm{Q,low}})) \tag{8}$$

denotes the intermediate representation. $\boldsymbol{W}_{\mathrm{I,low}} \in \mathbb{R}^{d_{\mathrm{low}} \times d_{\mathrm{low}}}$, $\boldsymbol{W}_{\mathrm{Q,low}} \in \mathbb{R}^{d_{\mathrm{low}} \times d_{\mathrm{low}}}$ and $\boldsymbol{W}_{\mathrm{A,low}} \in \mathbb{R}^{d_{\mathrm{low}} \times d_{\mathrm{low}}}$ are weight matrices, $\boldsymbol{b}_{\mathrm{Q,low}} \in \mathbb{R}^{d_{\mathrm{low}}}$ and $\boldsymbol{b}_{\mathrm{A,low}} \in \mathbb{R}^{d_{\mathrm{low}}}$ are biases. Finally, we aggregate $\boldsymbol{q}_{\mathrm{low}}$ with $\boldsymbol{v}_{\mathrm{KG}}$ based on the low-level attention values $\boldsymbol{\alpha}_{\mathrm{low}} \in \mathbb{R}^{d_{\mathrm{low}} \times m}$ to get state representation $\boldsymbol{v}_t \in \mathbb{R}^{d_{\mathrm{low}}}$:

$$\boldsymbol{v}_t = \boldsymbol{q}_{\mathrm{low}} + \sum_{i}^{m} \boldsymbol{\alpha}_{\mathrm{low},i} \odot \boldsymbol{v}_{\mathrm{KG},i} \tag{9}$$

### 4.4   Action selection and training

**Action selection.** Given the state representation $\boldsymbol{v}_t$, the action selection can be performed via methods such as template-based scoring [20] and recurrent decoding [3]. In this work, we use the recurrent decoding method to select actions via two GRUs. We first use a template-GRU to predict a template $\boldsymbol{u} \in \mathcal{T}$ based on $\boldsymbol{v}_t$, where $\mathcal{T}$ denotes the template set. Next, we recurrently execute an object-GRU for $k$ steps to decode objects $\{\boldsymbol{p}_i, i \in [1, ..., k]\}$ from the object set $\mathcal{P}$, which is the intersection of the vocabulary set $\mathcal{V}$ and the set of the objects appeared in $\boldsymbol{o}_{\mathrm{KG,full}}$. The probability of an object $\boldsymbol{p}_i$ is conditioned on both $\boldsymbol{v}_t$ and the prediction of the last step (i.e., $\boldsymbol{u}$ or $\boldsymbol{p}_{t-1}$). Finally, the template and objects are combined as action $\boldsymbol{a}_t$.

**Training.** Our model SHA-KG is trained via the Advantage Actor Critic (A2C) method [37] with a supervised auxiliary task "valid action prediction" [3]. We provide details in the supplementary material.

## 5   Experiments

We evaluate our method on a set of man-made games in Jericho game suite [20]. We conduct experiments to validate the effectiveness of our sub-graph division and stacked hierarchical attention, and interpret the reasoning and decision making processes.

### 5.1   Baselines

We use KG-A2C [3] as the building backbone of SHA-KG. Following baselines are considered:

- NAIL [21]: a general agent with hand-crafted rules and pre-trained language models.
- DRRN [20, 22]: a choice-based agent that selects actions from a valid action set.
- TDQN [20]: a parser-based agent with template-based action space.
- KG-A2C [3]: an extension of TDQN with KGs and valid action predictions.

### 5.2   Experimental setup

**Graph construction.**   The triples for constructing the KG are extracted via Stanford's Open Information Extraction (OpenIE) [6] and two additional rules used in [3]: 1) For interactive objects detected in the current observation, those within the inventory are linked to node "you", while others are linked to the current room. 2) The room connectivity is inferred from navigational actions. Regarding the sub-graph division, we define four graph types: $\boldsymbol{o}_{\mathrm{KG,1}}$ records the connectivity of visited rooms, $\boldsymbol{o}_{\mathrm{KG,2}}$ represents the objects within the current room, $\boldsymbol{o}_{\mathrm{KG,3}}$ represents the objects within the inventory, and $\boldsymbol{o}_{\mathrm{KG,4}}$ is the graph without any connection to "you". $\boldsymbol{o}_{\mathrm{KG,2}}$ and $\boldsymbol{o}_{\mathrm{KG,3}}$ contain the present information only, while $\boldsymbol{o}_{\mathrm{KG,1}}$ and $\boldsymbol{o}_{\mathrm{KG,4}}$ contain both the current and historical information. Besides pre-defined rules, the graph partitioning process can be implemented via automatic methods, which we leave as future work.

Table 1: Raw scores of SHA-KG and baselines. For the baselines, we use the results reported in their original paper [3, 20] except "reverb" and "tryst205" in KG-A2C, which are not reported. $|\mathcal{T}|$ and $|\mathcal{V}|$ denote the size of template set and vocabulary set. **MaxR** denotes the maximum possible score (collected based on walkthrough without step limit). All results are averaged over five independent runs.

| Game | $|\mathcal{T}|$ | $|\mathcal{V}|$ | NAIL | DRRN | TDQN | KG-A2C | SHA-KG (Ours) | MaxR |
|---|---|---|---|---|---|---|---|---|
| acorncourt | 151 | 343 | 0 | **10** | 1.6 | 0.3 | 1.6 | 30 |
| balances | 156 | 452 | **10** | **10** | 4.8 | **10.0** | **10.0** | 51 |
| detective | 197 | 344 | 136.9 | 197.8 | 169 | 207.9 | **308.0** | 360 |
| dragon | 177 | 1049 | **0.6** | -3.5 | -5.3 | 0 | 0.2 | 25 |
| enchanter | 290 | 722 | 0 | **20.0** | 8.6 | 12.1 | **20.0** | 400 |
| inhumane | 141 | 409 | 0.6 | 0 | 0.7 | 3 | **5.4** | 300 |
| jewel | 161 | 657 | 1.6 | 1.6 | 0 | **1.8** | **1.8** | 90 |
| library | 173 | 510 | 0.9 | 17 | 6.3 | 14.3 | 15.8 | 30 |
| ludicorp | 187 | 503 | 8.4 | 13.8 | 6 | **17.8** | **17.8** | 150 |
| pentari | 155 | 472 | 0 | 27.2 | 17.4 | 50.7 | **51.3** | 70 |
| reverb | 183 | 526 | 0 | 8.2 | 0.3 | 7.4 | **10.6** | 50 |
| sorcerer | 288 | 1013 | 5 | 20.8 | 5 | 5.8 | **29.4** | 400 |
| spellbrkr | 333 | 844 | **40** | 37.8 | 18.7 | 21.3 | **40.0** | 600 |
| spirit | 169 | 1112 | 1 | 0.8 | 0.6 | 1.3 | **3.8** | 250 |
| temple | 175 | 622 | 7.3 | 7.4 | **7.9** | 7.6 | **7.9** | 35 |
| tryst205 | 197 | 871 | 2 | **9.6** | 0 | 6.7 | 6.9 | 350 |
| zenon | 149 | 401 | 0 | 0 | 0 | **3.9** | **3.9** | 350 |
| zork1 | 237 | 697 | 10.3 | 32.6 | 9.9 | 34 | **34.5** | 350 |
| zork3 | 214 | 564 | **1.8** | 0.5 | 0 | 0.1 | 0.7 | 7 |
| ztuu | 186 | 607 | 0 | 21.6 | 4.9 | 9.2 | **25.2** | 100 |

**Training implementation.** We follow the hyper-parameter setting of KG-A2C [3] except that we reduce the node embedding dimension in GATs from 50 to 25 to reduce GPU cost. We set $d_{high}$ as 100, and $d_{low}$ as 50. For both SHA-KG and KG-A2C, the graph mask for action selection is constructed from $o_{KG,full}$. We denote an action as "valid" if it does not lead to meaningless feedback in a state (e.g., "Nothing happens"). An episode will be terminated after 100 valid steps or game over / victory. For each game, an individual agent is trained for $10^6$ interaction steps. The training data is collected from 32 environments in parallel. An optimization step is performed per 8 interaction steps via the Adam optimizer with the learning rate 0.003. All baselines follow their original implementations [3, 20, 21]. To compare with the baselines, we report the average raw score over the last 100 finished episodes during training. We also report the learning curve during ablation study. All the quantitative results are averaged over five independent runs.

## 5.3 Overall performance

Table 1 shows the performance of SHA-KG and baselines in 20 games. The proposed SHA-KG achieves the new state-of-the-art results in 8 games and is equivalent to the current best baselines in another 7 games. Comparing the three agents using a template-based action space (i.e., SHA-KG, TDQN, KG-A2C), SHA-KG obtains equal or better performance than TDQN and KG-A2C in all of the games, showing the effectiveness of introducing reasoning. The best result of the other 5 games is still achieved by NAIL and DRRN, which apply additional rules and assumptions. NAIL follows nearly fixed rules (e.g., interacting with all the objects, then exploring another room). Though this design principle may be useful for some particular games (e.g., "dragon" and "zork3"), it lacks flexibility so that it performs worse than all learning-based agents in most of the games. DRRN largely reduces the difficulty by selecting actions from the set of admissible actions only. For example, to obtain the first reward "+10" in "acorncourt", the agent has to select a high proportion of complex actions, in which case the assumption of an admissible action set is a large advantage. KG-A2C and our SHA-KG relax this assumption but the action set is as large as $\mathcal{O}(\mathcal{TP}^2)$, making them infeasible to achieve the first reward stably. However, the reasoning ability still brings improvement compared with the backbone model. While KG-A2C shows worse performance than DRRN in 9 games, SHA-KG outperforms DRRN in 6 of these games, and shows closer scores in other 3 games.

## 5.4 Ablation study

We perform ablation studies to validate the contributions of different components. We first compare SHA-KG with its three variants with different attention modules:

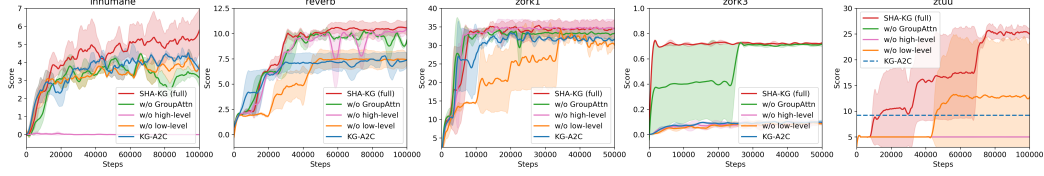

Figure 3: Learning curves of models with different attention modules. The dashed line in "ztuu" denotes KG-A2C's result reported in [3]. The shaded regions indicate standard deviations.

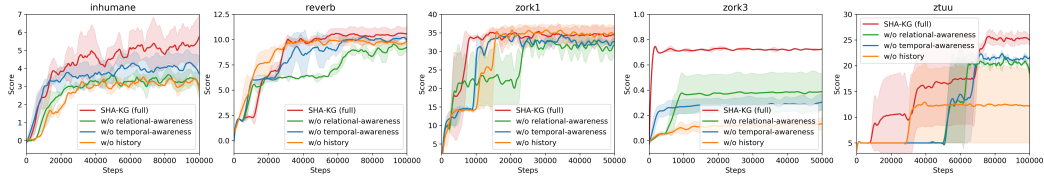

Figure 4: Learning curves of SHA-KG with different types of sub-graphs.

- "w/o GroupAttn": applies single attention value for each channel, i.e., $\boldsymbol{\alpha}_{\text{high}} \in \mathbb{R}^4$, $\boldsymbol{\alpha}_{\text{low}} \in \mathbb{R}^4$.

- "w/o high-level": constructs the initial query vector from $\boldsymbol{v}_{\text{text}}$ and $\boldsymbol{v}_{\text{score}}$, then computes attention values across the sub-graphs. The full KG is not used.

- "w/o low-level": constructs the initial query vector from $\boldsymbol{v}_{\text{KG, full}}$ and $\boldsymbol{v}_{\text{score}}$, then computes attention values across the textual components. The sub-graph division is not used.

Fig. 3 shows the learning curves of 5 games, where following observations can be made: 1) The full model SHA-KG shows similar or better performance than all variants in all cases. Our two-level attention mechanism provides an effective and explainable way to refine information. The first level of hierarchy tells the agent which part of the textual information should be focused on. Based on the output of this level, the second level of hierarchy informs the agent which part of a knowledge graph should be targeting at. 2) The variant "w/o high-level", which considers sub-graphs only but not the full graph (i.e., the pink curve), works for some games (e.g., "reverb" and "zork1") but fails for the others. For the games where the variant is able to achieve the best score, the sample efficiency is also improved, which means that this variant requires fewer interaction steps to achieve the best score. 3) For the variant "w/o low-level" that considers the full KG only but not sub-graphs (i.e., the orange curve), the sample efficiency is a bit low in some games (e.g., "zork1"). 4) The variant "w/o GroupAttn" generally shows a similar performance curve to SHA-KG by considering both the full KG and sub-graphs. However, the performance gap between "w/o GroupAttn" and SHA-KG demonstrates the effectiveness of computing multiple groups of attention values along the channels, which allows SHA-KG to capture more fine-grained information from textual components (in high level) and sub-graphs (in low level). Overall, the learning curves demonstrate that the sub-graph division and stacked hierarchical attention provide complementary contributions to the performance improvement of SHA-KG.

Regarding the contributions of different types of sub-graphs, we further design three variants with different graph partitioning strategies:

- "w/o relational-awareness" combines $\boldsymbol{o}_{\text{KG},2}$ (room objects) and $\boldsymbol{o}_{\text{KG},3}$ (collected objects).

- "w/o temporal-awareness" combines $\boldsymbol{o}_{\text{KG},4}$ with $\boldsymbol{o}_{\text{KG},2}$ and $\boldsymbol{o}_{\text{KG},3}$, respectively.

- "w/o history" removes all historical information.

Fig. 4 shows the results and indicates that the effect of different types of awareness varies with respect to the games. No simple conclusion can be made regarding which type of awareness contributes the most to the final performance (e.g., "w/o relational-awareness" and "w/o temporal-awareness" behave differently in "zork1" and "zork3"). However, considering them collectively and learning to balance their importance lead to the improved performance of our method.

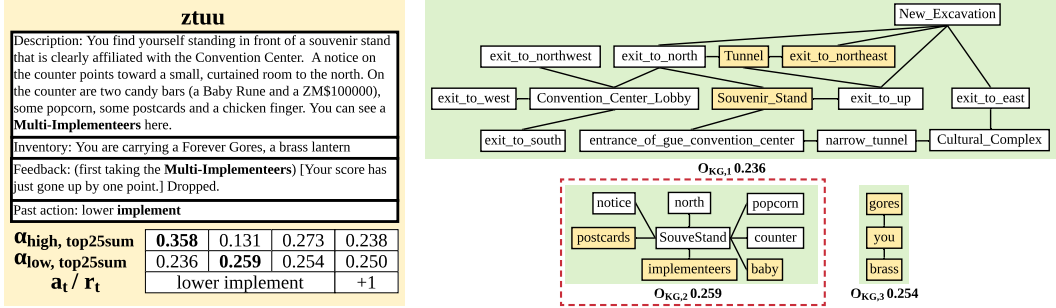

**ztuu**

Description: You find yourself standing in front of a souvenir stand that is clearly affiliated with the Convention Center. A notice on the counter points toward a small, curtained room to the north. On the counter are two candy bars (a Baby Rune and a ZM$100000), some popcorn, some postcards and a chicken finger. You can see a **Multi-Implementeers** here.

Inventory: You are carrying a Forever Gores, a brass lantern

Feedback: (first taking the **Multi-Implementeers**) [Your score has just gone up by one point.] Dropped.

Past action: lower **implement**

| | | | | |
|---|---|---|---|---|
| $\alpha_{\text{high, top25sum}}$ | **0.358** | 0.131 | 0.273 | 0.238 |
| $\alpha_{\text{low, top25sum}}$ | 0.236 | **0.259** | 0.254 | 0.250 |
| $a_t$ / $r_t$ | lower implement | | | +1 |

$O_{\text{KG},1}$ 0.236    $O_{\text{KG},2}$ 0.259    $O_{\text{KG},3}$ 0.254

Figure 5: Illustration of the reasoning and decision making processes of the game "ztuu". Left: $\alpha_{\text{high,top25sum}}$ denotes the high-level attention for $o_{\text{text,desc}}$, $o_{\text{text,inv}}$, $o_{\text{text,feed}}$ and $a_{\text{t-1}}$. $\alpha_{\text{low,top25sum}}$ denotes the low-level attention for $o_{\text{KG, 1}}$, $o_{\text{KG, 2}}$, $o_{\text{KG, 3}}$ and $o_{\text{KG, 3}}$. The subscript "top25sum" denotes the sum of 25 largest attention values within a channel (textual observation or sub-graph). $a_t$ is the selected action, and $r_t$ is the reward. Right: the extracted sub-graphs ($o_{\text{KG,4}}$ is omitted due to space limit). The sub-graph with the highest graph-level attention (by SHA) is in the red dashed box. In each sub-graph, the top 3 nodes with the highest attention (by GATs) are highlighted in yellow.

## 5.5 Interpretability

We interpret the reasoning and decision making process by examining the attentive focus. Although there are controversial points about whether attention values are explainable when being applied to text [25, 51], we believe our attention mechanism is beneficial for interpretability. Compared to word-level attention, the stacked hierarchical attention is conducted in a manner more similar to the region/channel-level attention mechanism, which has been proved to providing interpretability in vision tasks, especially visual-based RL tasks [18, 40] As multiple groups of attention values are computed (e.g., each sub-graph is associated with $d_{\text{low}}$ attention values), we aggregate them to facilitate explaining. Specifically, for each channel (e.g., textual component or sub-graph), we sum the top 25 largest values as its attention value, then perform softmax operation across the channels [4]. We denote the obtained attention values as $\alpha_{\text{high, top25sum}}$ for the high level, which correspond to $\langle o_{\text{text, desc}}, o_{\text{text, inv}}, o_{\text{text, feed}}, a_{\text{t-1}}\rangle$, and $\alpha_{\text{low, top25sum}}$ for the low level, which correspond to $\langle o_{\text{KG,1}}, o_{\text{KG,2}}, o_{\text{KG,3}}, o_{\text{KG,4}}\rangle$. Fig. 5 (left) shows a decision making example of the game "ztuu". In the high level (textual components), the agent focuses mostly on the description $o_{\text{text, desc}}$, and then the feedback $o_{\text{text, feed}}$, which is followed by the last action $a_{\text{t-1}}$. All of the three text components contain "implement". In the low level (sub-graphs), the sub-graph of objects in the current room ($o_{\text{KG,2}}$) has the highest attention. Combining both two levels of attention, the agent finally selects the action "lower implement", which receives a positive reward. It shows that the reasoning process enables the agent to select actions leading to positive rewards. Although the GATs in our work are mainly used for obtaining initial graph embeddings instead of assigning attention values, we also visualize the node-level attention within sub-graphs to help understand the reasoning process. Fig. 5 (right) shows three extracted sub-graphs. The digit under the sub-graph denotes graph-level attention. Since the $o_{\text{KG,2}}$ has the highest attention, the agent will focus more on objects it contains. In each sub-graph, nodes with top-3 highest attention (by GATs) are highlighted in yellow. Such node-level attention helps to further constrain (softly) the objects in $o_{\text{KG,2}}$ to derive actions. We conclude that our SHA helps the agent to use information efficiently for taking actions.

## 6 Conclusion

In this paper, we have studied empowering RL for text-based games with reasoning by exploiting knowledge graphs. We conducted sub-graph division to explicitly introduce relational-awareness and temporal-awareness. Then we designed a stacked hierarchical attention mechanism to obtain effective state representation from multi-modal inputs. Besides obtaining favorable experimental results in a wide range of man-made games, the sub-graph division and attention mechanism enable us to better interpret the reasoning and decision making processes of RL agents.

## Broader Impact

The high-level goal of this work is to bridge artificial intelligence with human intelligence, cognition and language learning. Researchers in reinforcement learning will benefit from this work by the appropriate use of knowledge graphs. By recording and organizing the information in a structural way, the difficulty of learning can be largely reduced. Besides, KGs can be used to conduct reasoning to interpret the decision making process. Researchers in multi-modal learning will also benefit from the attention mechanism proposed in this work. Although the stacked hierarchical attention is used to aggregate text representation and graph representation, it can also be extended to other forms of inputs such as visual and audio signals. Our work can also be served as an initial study before conducting experiments in real life and with animal / human participants, since it's performed in simulated systems and there's no safety consideration.

Regarding the ethical implications, although currently our work is conducted in games, where language commands are constrained in limited action spaces, for more practical applications this system will be deployed with a richer corpus. From the perspective of language generation, the inappropriate use of generated language commands should be seriously taken into consideration. Another concern lies in the unintended behavior and decision making process, which may lead to dangerous conditions in real world applications. Although we try to improve interpretability through reasoning, there's still a long way to go to make human-AI interaction in a safe and reasonable way.

## Acknowledgements

We would like to thank the anonymous reviewers for helpful feedback. This research was partly supported by ARC Discovery Project DP180100966.

## Footnotes

[2]Our code is available at `https://github.com/YunqiuXu/SHA-KG`.

[3]As discussed in 4.1, $c$ is 4 in this work.

[4]We also conducted other aggregation methods such as "top10, sum", "top25, mean" and "all, sum", among which we found that "top25, sum" can best interpret the processes. See the supplementary material.

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
