[Supplementary Material]

# SUPPLEMENTARY MATERIAL: Deep Reinforcement Learning with Stacked Hierarchical Attention for Text-based Games

In the supplementary material, we describe the training details, examples of game interface and interactions used in the paper.

## A   Training details

We train our model using the Advantage Actor Critic (A2C) method [37] across valid actions. Given $s_t$, the valid action set $\text{Valid}(s_t)$ consists of actions not leading to meaningless feedback (e.g., "Nothing happens"). Function to obtain the valid action set is provided by Jericho [20]. We denote all learnable parameters of our model as $\boldsymbol{\theta}_t$. After obtaining the state representation vector $\boldsymbol{v}_t$, we use a critic network to estimate the value $V(\boldsymbol{v}_t)$ and treat decoders for templates and objects as different policies: $\boldsymbol{\pi}_{\mathbb{T}}$ and $\boldsymbol{\pi}_{\mathbb{O}}$. We first compute the advantage $A(\boldsymbol{a}_t, \boldsymbol{v}_t)$, then compute the policy loss $\mathcal{L}_{\pi}(\boldsymbol{v}_t, \boldsymbol{a}_t; \boldsymbol{\theta}_t)$ and the critic loss $\mathcal{L}_{\text{critic}}(\boldsymbol{v}_t, \boldsymbol{a}_t; \boldsymbol{\theta}_t)$ according to the gradient. We also add an entropy loss $\mathcal{L}_{\mathbb{E}}(\boldsymbol{v}_t, \boldsymbol{a}_t; \boldsymbol{\theta}_t)$ to encourage diversity.

Q-value function:

$$Q(\boldsymbol{v}_t, \boldsymbol{a}_t) = \boldsymbol{r}_t + \gamma V(\boldsymbol{v}_{t+1}) \tag{10}$$

Advantage function:

$$A(\boldsymbol{v}_t, \boldsymbol{a}_t) = Q(\boldsymbol{v}_t, \boldsymbol{a}_t) - V(\boldsymbol{v}_t) \tag{11}$$

Actor loss:

$$\mathcal{L}_{\pi}(\boldsymbol{v}_t, \boldsymbol{a}_t; \boldsymbol{\theta}_t) = \mathbb{E}\left[\left(-\log \pi_{\mathbb{T}}(\boldsymbol{u}|\boldsymbol{v}_t; \boldsymbol{\theta}_t) - \sum_{i=1}^{n} \log \pi_{\mathbb{O}_i}(\boldsymbol{p}_i|\boldsymbol{v}_t, \boldsymbol{u}, ..., \boldsymbol{p}_{i-1}; \boldsymbol{\theta}_t)\right) A(\boldsymbol{v}_t, \boldsymbol{a}_t)\right] \tag{12}$$

Critic loss:

$$\mathcal{L}_{\text{critic}}(\boldsymbol{v}_t, \boldsymbol{a}_t; \boldsymbol{\theta}_t) = \mathbb{E}[\boldsymbol{r}_t + \gamma V(\boldsymbol{v}_{t+1}) - V(\boldsymbol{v}_t; \boldsymbol{\theta}_t)] \tag{13}$$

Entropy loss:

$$\mathcal{L}_{\mathbb{E}}(\boldsymbol{v}_t, \boldsymbol{a}_t; \boldsymbol{\theta}_t) = \sum_{\boldsymbol{a} \in \text{Valid}(\boldsymbol{s}_t)} P(\boldsymbol{a}|\boldsymbol{v}_t) \log P(\boldsymbol{a}|\boldsymbol{v}_t) \tag{14}$$

Similar to KG-A2C [3], a supervised auxiliary task "valid action prediction" is introduced to assist RL training. We first extract the valid template set $\mathcal{T}_{\text{valid}}(s_t) = \{\boldsymbol{\tau}_0, \boldsymbol{\tau}_1, ..., \boldsymbol{\tau}_N\}$ from the valid action set $\text{Valid}(\boldsymbol{s}_t)$, and valid object set $\mathcal{O}_{\text{valid}}(s_t) = \{\boldsymbol{o}_0, \boldsymbol{o}_1, ..., \boldsymbol{o}_M\}$ from the full knowledge graph $\boldsymbol{o}_{t,\text{KG}}$. Then we treat "whether the predicted template / objects is valid" as binary classification thus introduce two cross entropy loss functions: $\mathcal{L}_{\mathbb{T}}(\boldsymbol{v}_t, \boldsymbol{a}_t; \boldsymbol{\theta}_t)$ for the template and $\mathcal{L}_{\mathbb{O}}(\boldsymbol{v}_t, \boldsymbol{a}_t; \boldsymbol{\theta}_t)$ for the objects.

Template loss:

$$\mathcal{L}_{\mathbb{T}}(\boldsymbol{v}_t, \boldsymbol{a}_t; \boldsymbol{\theta}_t) = \frac{1}{N} \sum_{i=1}^{N} \left(\boldsymbol{y}_{\tau_i} \log \pi_{\mathbb{T}}(\boldsymbol{\tau}_i|\boldsymbol{v}_t) + (1 - \boldsymbol{y}_{\tau_i})(1 - \log \pi_{\mathbb{T}}(\boldsymbol{\tau}_i|\boldsymbol{v}_t))\right) \tag{15}$$

Object loss:

$$\mathcal{L}_{\mathbb{O}}(\boldsymbol{v}_t, \boldsymbol{a}_t; \boldsymbol{\theta}_t) = \sum_{j=1}^{k} \frac{1}{M} \sum_{i=1}^{M} \left(\boldsymbol{y}_{o_i} \log \pi_{\mathbb{O}_j}(\boldsymbol{o}_i|\boldsymbol{v}_t) + (1 - \boldsymbol{y}_{o_i})(1 - \log \pi_{\mathbb{O}_j}(\boldsymbol{o}_i|\boldsymbol{v}_t))\right) \tag{16}$$

where $k$ denotes the number of object decoding steps (there are at most $k$ objects in an action). We have $\boldsymbol{y}_{\tau_i} = 1$ if the template $\boldsymbol{\tau}_i \in \mathcal{T}_{\text{valid}}(s_t)$, else 0. Similarly, we have $\boldsymbol{y}_{o_i} = 1$ if the object $\boldsymbol{o}_i \in \mathcal{O}_{\text{valid}}(s_t)$, else 0. $\boldsymbol{\theta}_t$ will be optimized jointly with the total loss $\mathcal{L}_{\text{total}}(\boldsymbol{v}_t, \boldsymbol{a}_t; \boldsymbol{\theta}_t)$, which is the weighted sum of the above five losses:

$$\mathcal{L}_{\text{total}}(\boldsymbol{v}_t, \boldsymbol{a}_t; \boldsymbol{\theta}_t) = \mathcal{L}_{\pi} + \boldsymbol{\lambda}_{\text{critic}}\mathcal{L}_{\text{critic}} + \boldsymbol{\lambda}_{\mathbb{E}}\mathcal{L}_{\mathbb{E}} + \boldsymbol{\lambda}_{\mathbb{T}}\mathcal{L}_{\mathbb{T}} + \boldsymbol{\lambda}_{\mathbb{O}}\mathcal{L}_{\mathbb{O}} \tag{17}$$

where $\boldsymbol{\lambda}_{\text{critic}}$, $\boldsymbol{\lambda}_{\mathbb{E}}$, $\boldsymbol{\lambda}_{\mathbb{T}}$ and $\boldsymbol{\lambda}_{\mathbb{O}}$ are coefficients.

Figure 6: Interface of the game "ztuu". Action (blue) is a textual command, and raw observation (black) is the textual feedback of previous actions.

# B  Interface

Fig. 6 shows an example of the raw interface of the game "ztuu", where raw textual observations contain only the feedback of taking an action.

# C  Interaction examples

In this section, we show the first 15 interaction steps of two games: "zork1" and "ztuu". Each interaction step consists of the current textual observation, triplets extracted from the current textual observation, high-level and low-level attention values obtained via different aggregation methods, chosen actions and rewards.

## C.1  zork1

```
———— ===== Step 1 ===== ————
===== 1. Textual obs:
o_desc: West of House. You are standing in an open field west of a
    white house, with a boarded front door. There is a small
    mailbox here.
o_inv: You are empty−handed.
o_feed: Copyright (c) 1981, 1982, 1983 Infocom, Inc. All rights
    reserved. ZORK is a registered trademark of Infocom, Inc.
    Revision 88 / Serial number 840726 West of House You are
    standing in an open field west of a white house, with a
    boarded front door. There is a small mailbox here.
a_past: look
===== 2. Newly extracted triplets
[('West of House', 'has', 'exit to west'), ('all', 'in', 'west of
    white house with boarded front door'), ('door', 'in', 'west of
     white house with boarded front door'), ('house', 'in', 'west
    of white house with boarded front door'), ('mailbox', 'in', '
    west of white house with boarded front door'), ('you', 'in', '
    field'), ('you', 'in', 'open field'), ('you', 'in', 'west'), (
    'you', 'in', 'west of house'), ('you', 'in', 'west of house
    with boarded front door'), ('you', 'in', 'west of house with
    front door'), ('you', 'in', 'west of white house'), ('you', '
    in', 'west of white house with boarded front door'), ('you', '
    in', 'west of white house with front door'), ('you', 'in', '
    west with boarded front door'), ('you', 'in', 'west with front
     door')]
===== 3. Attention values:
———— attH: o_desc, o_inv, o_feed, a_past
attH_max        : ['0.262', '0.262', '0.262', '0.213']
attH_mean       : ['0.320', '0.205', '0.267', '0.208']
attH_sum        : ['1.000', '0.000', '0.000', '0.000']
attH_top10_mean : ['0.308', '0.202', '0.308', '0.182']
attH_top10_sum  : ['0.497', '0.007', '0.493', '0.003']
```

```
attH_top25_mean: ['0.342', '0.169', '0.316', '0.172']
attH_top25_sum : ['0.881', '0.000', '0.119', '0.000']
attH_top50_mean: ['0.361', '0.174', '0.286', '0.179']
attH_top50_sum : ['1.000', '0.000', '0.000', '0.000']
——— attL: connectivity, item_in_room, item_in_inv, history
attL_max       : ['0.250', '0.250', '0.250', '0.250']
attL_mean      : ['0.250', '0.250', '0.250', '0.250']
attL_sum       : ['0.251', '0.250', '0.251', '0.248']
attL_top10_mean: ['0.250', '0.250', '0.250', '0.250']
attL_top10_sum : ['0.249', '0.251', '0.248', '0.252']
attL_top25_mean: ['0.250', '0.250', '0.250', '0.250']
attL_top25_sum : ['0.249', '0.251', '0.248', '0.252']
attL_top50_mean: ['0.250', '0.250', '0.250', '0.250']
attL_top50_sum : ['0.251', '0.250', '0.251', '0.248']
===== 4. Chosen action and reward
Action: west
Reward: 0|Score: 0

——— ===== Step 2 ===== ———
===== 1. Textual obs:
o_desc: Forest This is a forest, with trees in all directions. To
    the east, there appears to be sunlight.
o_inv: You are empty-handed.
o_feed: Forest This is a forest, with trees in all directions. To
    the east, there appears to be sunlight.
a_past: west
===== 2. Newly extracted triplets
[('Forest', 'has', 'exit to east'), ('all', 'in', 'Forest'), ('
    forest', 'is with', 'trees in directions'), ('forest this', '
    is', 'forest'), ('forest this', 'is forest with', 'trees in
    directions'), ('trees', 'in', 'Forest'), ('trees', 'is in', '
    directions'), ('west of white house with boarded front door',
    'west of', 'Forest')]
===== 3. Attention values:
——— attH: o_desc, o_inv, o_feed, a_past
attH_max       : ['0.264', '0.265', '0.265', '0.206']
attH_mean      : ['0.317', '0.207', '0.264', '0.213']
attH_sum       : ['1.000', '0.000', '0.000', '0.000']
attH_top10_mean: ['0.304', '0.216', '0.304', '0.176']
attH_top10_sum : ['0.491', '0.016', '0.491', '0.002']
attH_top25_mean: ['0.342', '0.174', '0.308', '0.175']
attH_top25_sum : ['0.930', '0.000', '0.070', '0.000']
attH_top50_mean: ['0.361', '0.176', '0.279', '0.184']
attH_top50_sum : ['1.000', '0.000', '0.000', '0.000']
——— attL: connectivity, item_in_room, item_in_inv, history
attL_max       : ['0.250', '0.250', '0.250', '0.250']
attL_mean      : ['0.250', '0.250', '0.250', '0.250']
attL_sum       : ['0.252', '0.250', '0.249', '0.249']
attL_top10_mean: ['0.250', '0.250', '0.250', '0.250']
attL_top10_sum : ['0.250', '0.249', '0.248', '0.253']
attL_top25_mean: ['0.250', '0.250', '0.250', '0.250']
attL_top25_sum : ['0.250', '0.249', '0.247', '0.253']
attL_top50_mean: ['0.250', '0.250', '0.250', '0.250']
attL_top50_sum : ['0.252', '0.250', '0.249', '0.249']
===== 4. Chosen action and reward
Action: south
Reward: 0|Score: 0

——— ===== Step 3 ===== ———
```

```
===== 1. Textual obs:
o_desc: Forest This is a dimly lit forest, with large trees all
    around.
o_inv: You are empty-handed.
o_feed: Forest This is a dimly lit forest, with large trees all
    around. You hear in the distance the chirping of a song bird.
a_past: south
===== 2. Newly extracted triplets
[('all', 'in', 'Forest'), ('forest this', 'is', 'dimly lit forest'
    ), ('forest this', 'is', 'forest'), ('forest this', 'is', 'lit
     forest'), ('forest this', 'is dimly lit forest with', 'large
    trees'), ('forest this', 'is dimly lit forest with', 'large
    trees around'), ('forest this', 'is dimly lit forest with', '
    trees'), ('forest this', 'is dimly lit forest with', 'trees
    around'), ('forest this', 'is forest with', 'large trees'), ('
    forest this', 'is forest with', 'large trees around'), ('
    forest this', 'is forest with', 'trees'), ('forest this', 'is
    forest with', 'trees around'), ('forest this', 'is lit forest
    with', 'large trees'), ('forest this', 'is lit forest with', '
    large trees around'), ('forest this', 'is lit forest with', '
    trees'), ('forest this', 'is lit forest with', 'trees around')
    , ('large', 'in', 'Forest'), ('lit forest', 'is with', 'large
    trees around'), ('this', 'is', 'lit'), ('trees', 'in', 'Forest
    '), ('west of white house with boarded front door', 'south of'
    , 'Forest'), ('you', 'hear', 'chirping'), ('you', 'hear', '
    chirping of song bird'), ('you', 'hear chirping in', 'distance
    ')]
===== 3. Attention values:
------- attH: o_desc, o_inv, o_feed, a_past
attH_max        : ['0.261', '0.261', '0.260', '0.218']
attH_mean       : ['0.309', '0.208', '0.255', '0.228']
attH_sum        : ['1.000', '0.000', '0.000', '0.000']
attH_top10_mean: ['0.294', '0.215', '0.292', '0.199']
attH_top10_sum : ['0.502', '0.022', '0.465', '0.010']
attH_top25_mean: ['0.334', '0.176', '0.290', '0.199']
attH_top25_sum : ['0.973', '0.000', '0.027', '0.000']
attH_top50_mean: ['0.350', '0.179', '0.263', '0.208']
attH_top50_sum : ['1.000', '0.000', '0.000', '0.000']
------- attL: connectivity, item_in_room, item_in_inv, history
attL_max        : ['0.250', '0.250', '0.250', '0.251']
attL_mean       : ['0.250', '0.250', '0.250', '0.250']
attL_sum        : ['0.252', '0.251', '0.250', '0.248']
attL_top10_mean: ['0.250', '0.250', '0.250', '0.250']
attL_top10_sum : ['0.249', '0.249', '0.247', '0.255']
attL_top25_mean: ['0.250', '0.250', '0.250', '0.250']
attL_top25_sum : ['0.250', '0.249', '0.246', '0.255']
attL_top50_mean: ['0.250', '0.250', '0.250', '0.250']
attL_top50_sum : ['0.252', '0.251', '0.250', '0.248']
===== 4. Chosen action and reward
Action: south
Reward: 0|Score: 0

------- ===== Step 4 ===== -------
===== 1. Textual obs:
o_desc: Forest This is a dimly lit forest, with large trees all
    around.
o_inv: You are empty-handed.
o_feed: Storm-tossed trees block your way.
a_past: south
```

```
===== 2. Newly extracted triplets
[('all', 'in', 'Forest'), ('large', 'in', 'Forest'), ('lit forest'
    , 'is with', 'large trees around'), ('this', 'is', 'lit'), ('
    trees', 'block', 'your way'), ('trees', 'in', 'Forest'), ('
    west of white house with boarded front door', 'south of', '
    Forest')]
===== 3. Attention values:
———— attH: o_desc, o_inv, o_feed, a_past
attH_max        : ['0.263', '0.264', '0.263', '0.210']
attH_mean       : ['0.315', '0.207', '0.253', '0.224']
attH_sum        : ['1.000', '0.000', '0.000', '0.000']
attH_top10_mean: ['0.297', '0.212', '0.297', '0.194']
attH_top10_sum : ['0.488', '0.017', '0.488', '0.007']
attH_top25_mean: ['0.338', '0.174', '0.294', '0.194']
attH_top25_sum : ['0.971', '0.000', '0.029', '0.000']
attH_top50_mean: ['0.357', '0.178', '0.262', '0.203']
attH_top50_sum : ['1.000', '0.000', '0.000', '0.000']
———— attL: connectivity, item_in_room, item_in_inv, history
attL_max        : ['0.250', '0.250', '0.250', '0.251']
attL_mean       : ['0.250', '0.250', '0.250', '0.250']
attL_sum        : ['0.252', '0.250', '0.250', '0.247']
attL_top10_mean: ['0.250', '0.250', '0.250', '0.250']
attL_top10_sum : ['0.250', '0.248', '0.247', '0.255']
attL_top25_mean: ['0.250', '0.250', '0.250', '0.250']
attL_top25_sum : ['0.252', '0.248', '0.246', '0.254']
attL_top50_mean: ['0.250', '0.250', '0.250', '0.250']
attL_top50_sum : ['0.252', '0.250', '0.250', '0.247']
===== 4. Chosen action and reward
Action: west
Reward: 0|Score: 0

———— ===== Step 5 ===== ————
===== 1. Textual obs:
o_desc: Forest This is a forest, with trees in all directions. To
    the east, there appears to be sunlight.
o_inv: You are empty-handed.
o_feed: Forest
a_past: west
===== 2. Newly extracted triplets
[('Forest', 'has', 'exit to east'), ('all', 'in', 'Forest'), ('
    forest', 'is with', 'trees in directions'), ('trees', 'in', '
    Forest'), ('trees', 'is in', 'directions'), ('west of white
    house with boarded front door', 'west of', 'Forest')]
===== 3. Attention values:
———— attH: o_desc, o_inv, o_feed, a_past
attH_max        : ['0.268', '0.267', '0.267', '0.198']
attH_mean       : ['0.325', '0.206', '0.259', '0.210']
attH_sum        : ['1.000', '0.000', '0.000', '0.000']
attH_top10_mean: ['0.308', '0.213', '0.304', '0.175']
attH_top10_sum : ['0.523', '0.013', '0.462', '0.002']
attH_top25_mean: ['0.349', '0.174', '0.302', '0.175']
attH_top25_sum : ['0.973', '0.000', '0.027', '0.000']
attH_top50_mean: ['0.372', '0.176', '0.271', '0.181']
attH_top50_sum : ['1.000', '0.000', '0.000', '0.000']
———— attL: connectivity, item_in_room, item_in_inv, history
attL_max        : ['0.250', '0.250', '0.250', '0.250']
attL_mean       : ['0.250', '0.250', '0.250', '0.250']
attL_sum        : ['0.251', '0.249', '0.249', '0.251']
attL_top10_mean: ['0.250', '0.250', '0.250', '0.250']
```

```
attL_top10_sum  : ['0.249', '0.249', '0.250', '0.252']
attL_top25_mean: ['0.250', '0.250', '0.250', '0.250']
attL_top25_sum  : ['0.250', '0.249', '0.249', '0.252']
attL_top50_mean: ['0.250', '0.250', '0.250', '0.250']
attL_top50_sum  : ['0.251', '0.249', '0.249', '0.251']
===== 4. Chosen action and reward
Action: east
Reward: 0| Score: 0

——— ===== Step 6 ===== ———
===== 1. Textual obs:
o_desc: Forest Path This is a path winding through a dimly lit
    forest. The path heads north-south here. One particularly
    large tree with some low branches stands at the edge of the
    path.
o_inv: You are empty-handed. You hear in the distance the chirping
     of a song bird.
o_feed: Forest Path This is a path winding through a dimly lit
    forest. The path heads north-south here. One particularly
    large tree with some low branches stands at the edge of the
    path.
a_past: east
===== 2. Newly extracted triplets
[('Forest Path', 'has', 'exit to north'), ('Forest Path', 'has', '
    exit to south'), ('all', 'in', 'Forest Path'), ('forest', 'in'
    , 'Forest Path'), ('one large tree', 'is with', 'low branches'
    ), ('path', 'in', 'Forest Path'), ('path', 'winding through',
    'dimly lit forest'), ('path', 'winding through', 'forest'), ('
    path', 'winding through', 'lit forest'), ('tree', 'in', '
    Forest Path'), ('west of white house with boarded front door',
    'east of', 'Forest Path')]
===== 3. Attention values:
——— attH: o_desc, o_inv, o_feed, a_past
attH_max        : ['0.254', '0.250', '0.255', '0.241']
attH_mean       : ['0.284', '0.223', '0.276', '0.217']
attH_sum        : ['0.932', '0.000', '0.068', '0.000']
attH_top10_mean: ['0.281', '0.231', '0.287', '0.201']
attH_top10_sum  : ['0.415', '0.059', '0.511', '0.015']
attH_top25_mean: ['0.302', '0.203', '0.313', '0.182']
attH_top25_sum  : ['0.294', '0.000', '0.706', '0.000']
attH_top50_mean: ['0.307', '0.203', '0.299', '0.191']
attH_top50_sum  : ['0.775', '0.000', '0.225', '0.000']
——— attL: connectivity, item_in_room, item_in_inv, history
attL_max        : ['0.250', '0.250', '0.250', '0.250']
attL_mean       : ['0.250', '0.250', '0.250', '0.250']
attL_sum        : ['0.251', '0.251', '0.251', '0.247']
attL_top10_mean: ['0.250', '0.250', '0.250', '0.250']
attL_top10_sum  : ['0.249', '0.251', '0.248', '0.252']
attL_top25_mean: ['0.250', '0.250', '0.250', '0.250']
attL_top25_sum  : ['0.249', '0.252', '0.247', '0.253']
attL_top50_mean: ['0.250', '0.250', '0.250', '0.250']
attL_top50_sum  : ['0.251', '0.251', '0.251', '0.247']
===== 4. Chosen action and reward
Action: up
Reward: 0| Score: 0

——— ===== Step 7 ===== ———
===== 1. Textual obs:
```

```
o_desc: Up a Tree You are about 10 feet above the ground nestled
    among some large branches. The nearest branch above you is
    above your reach. Beside you on the branch is a small bird's
    nest. In the bird's nest is a large egg encrusted with
    precious jewels, apparently scavenged by a childless songbird.
    The egg is covered with fine gold inlay, and ornamented in
    lapis lazuli and mother-of-pearl. Unlike most eggs, this one
    is hinged and closed with a delicate looking clasp. The egg
    appears extremely fragile.
o_inv: You are empty-handed.
o_feed: Up a Tree You are about 10 feet above the ground nestled
    among some large branches. The nearest branch above you is
    above your reach. Beside you on the branch is a small bird's
    nest. In the bird's nest is a large egg encrusted with
    precious jewels, apparently scavenged by a childless songbird.
    The egg is covered with fine gold inlay, and ornamented in
    lapis lazuli and mother-of-pearl. Unlike most eggs, this one
    is hinged and closed with a delicate looking clasp. The egg
    appears extremely fragile.
a_past: up
===== 2. Newly extracted triplets
[('all', 'in', 'up tree about 10 feet above ground nestled among
    large branches'), ('branch', 'in', 'up tree about 10 feet
    above ground nestled among large branches'), ('egg', 'in', 'up
     tree about 10 feet above ground nestled among large branches'
    ), ('ground', 'in', 'up tree about 10 feet above ground
    nestled among large branches'), ('nest', 'in', 'up tree about
    10 feet above ground nestled among large branches'), ('west of
     white house with boarded front door', 'up of', 'up tree about
     10 feet above ground nestled among large branches'), ('you',
    'in', 'about 10 feet'), ('you', 'in', 'about 10 feet above
    ground'), ('you', 'in', 'about 10 feet above ground nestled'),
     ('you', 'in', 'about 10 feet above ground nestled among
    branches'), ('you', 'in', 'about 10 feet above ground nestled
    among large branches'), ('you', 'in', 'about 10 feet nestled')
    , ('you', 'in', 'about 10 feet nestled among branches'), ('you
    ', 'in', 'about 10 feet nestled among large branches'), ('you'
    , 'in', 'up tree about 10 feet'), ('you', 'in', 'up tree about
     10 feet above ground'), ('you', 'in', 'up tree about 10 feet
    above ground nestled'), ('you', 'in', 'up tree about 10 feet
    above ground nestled among branches'), ('you', 'in', 'up tree
    about 10 feet above ground nestled among large branches'), ('
    you', 'in', 'up tree about 10 feet nestled'), ('you', 'in', '
    up tree about 10 feet nestled among branches'), ('you', 'in',
    'up tree about 10 feet nestled among large branches')]
===== 3. Attention values:
——— attH: o_desc, o_inv, o_feed, a_past
attH_max        : ['0.258', '0.253', '0.259', '0.231']
attH_mean       : ['0.263', '0.202', '0.331', '0.204']
attH_sum        : ['0.000', '0.000', '1.000', '0.000']
attH_top10_mean : ['0.311', '0.181', '0.320', '0.187']
attH_top10_sum  : ['0.422', '0.002', '0.574', '0.003']
attH_top25_mean : ['0.317', '0.162', '0.354', '0.168']
attH_top25_sum  : ['0.061', '0.000', '0.939', '0.000']
attH_top50_mean : ['0.280', '0.167', '0.381', '0.171']
attH_top50_sum  : ['0.000', '0.000', '1.000', '0.000']
——— attL: connectivity, item_in_room, item_in_inv, history
attL_max        : ['0.250', '0.250', '0.250', '0.250']
attL_mean       : ['0.250', '0.250', '0.250', '0.250']
```

```
attL_sum       : ['0.250', '0.251', '0.250', '0.248']
attL_top10_mean: ['0.250', '0.250', '0.250', '0.250']
attL_top10_sum : ['0.249', '0.251', '0.249', '0.251']
attL_top25_mean: ['0.250', '0.250', '0.250', '0.250']
attL_top25_sum : ['0.250', '0.251', '0.248', '0.251']
attL_top50_mean: ['0.250', '0.250', '0.250', '0.250']
attL_top50_sum : ['0.250', '0.251', '0.250', '0.248']
===== 4. Chosen action and reward
Action: take egg
Reward: 5|Score: 5

———— ===== Step 8 ===== ————
===== 1. Textual obs:
o_desc: Up a Tree You are about 10 feet above the ground nestled
    among some large branches. The nearest branch above you is
    above your reach. Beside you on the branch is a small bird's
    nest.
o_inv: You are carrying: A jewel−encrusted egg
o_feed: Taken.
a_past: take egg
===== 2. Newly extracted triplets
[('all', 'in', 'about 10 feet'), ('branch', 'in', 'about 10 feet')
    , ('egg', 'in', 'about 10 feet'), ('ground', 'in', 'about 10
    feet'), ('nest', 'in', 'about 10 feet'), ('you', 'have', ''),
    ('you', 'have', 'jewel−encrusted egg'), ('you', 'in', 'about
    10 feet')]
===== 3. Attention values:
———— attH: o_desc, o_inv, o_feed, a_past
attH_max       : ['0.263', '0.211', '0.263', '0.262']
attH_mean      : ['0.348', '0.201', '0.248', '0.203']
attH_sum       : ['1.000', '0.000', '0.000', '0.000']
attH_top10_mean: ['0.318', '0.184', '0.310', '0.188']
attH_top10_sum : ['0.557', '0.002', '0.438', '0.003']
attH_top25_mean: ['0.362', '0.169', '0.296', '0.173']
attH_top25_sum : ['0.993', '0.000', '0.007', '0.000']
attH_top50_mean: ['0.395', '0.173', '0.257', '0.175']
attH_top50_sum : ['1.000', '0.000', '0.000', '0.000']
———— attL: connectivity, item_in_room, item_in_inv, history
attL_max       : ['0.250', '0.249', '0.248', '0.253']
attL_mean      : ['0.250', '0.250', '0.250', '0.250']
attL_sum       : ['0.255', '0.250', '0.251', '0.244']
attL_top10_mean: ['0.250', '0.249', '0.249', '0.251']
attL_top10_sum : ['0.252', '0.245', '0.240', '0.263']
attL_top25_mean: ['0.250', '0.250', '0.249', '0.250']
attL_top25_sum : ['0.256', '0.245', '0.237', '0.262']
attL_top50_mean: ['0.250', '0.250', '0.250', '0.250']
attL_top50_sum : ['0.255', '0.250', '0.251', '0.244']
===== 4. Chosen action and reward
Action: put egg in egg
Reward: 0|Score: 5

———— ===== Step 9 ===== ————
===== 1. Textual obs:
o_desc: Up a Tree You are about 10 feet above the ground nestled
    among some large branches. The nearest branch above you is
    above your reach. Beside you on the branch is a small bird's
    nest.
o_inv: You are carrying: A jewel−encrusted egg You hear in the
    distance the chirping of a song bird.
```

```
o_feed : The jewel−encrusted egg isn't open.
a_past : put egg in egg
===== 2. Newly extracted triplets
[('all', 'in', 'about 10 feet'), ('branch', 'in', 'about 10 feet')
    , ('egg', 'in', 'about 10 feet'), ('ground', 'in', 'about 10
    feet'), ('nest', 'in', 'about 10 feet'), ('you', 'have', ''),
    ('you', 'have', 'hear in the distance the chirping of a song
    bird.'), ('you', 'have', 'jewel−encrusted egg'), ('you', 'in',
    'about 10 feet')]
===== 3. Attention values:
─────── attH: o_desc, o_inv, o_feed, a_past
attH_max        : ['0.252', '0.244', '0.253', '0.251']
attH_mean       : ['0.331', '0.213', '0.250', '0.206']
attH_sum        : ['1.000', '0.000', '0.000', '0.000']
attH_top10_mean: ['0.297', '0.208', '0.295', '0.200']
attH_top10_sum : ['0.503', '0.014', '0.474', '0.009']
attH_top25_mean: ['0.346', '0.189', '0.291', '0.174']
attH_top25_sum : ['0.987', '0.000', '0.013', '0.000']
attH_top50_mean: ['0.373', '0.191', '0.258', '0.178']
attH_top50_sum : ['1.000', '0.000', '0.000', '0.000']
─────── attL: connectivity, item_in_room, item_in_inv, history
attL_max        : ['0.250', '0.249', '0.249', '0.252']
attL_mean       : ['0.250', '0.250', '0.250', '0.250']
attL_sum        : ['0.252', '0.251', '0.250', '0.247']
attL_top10_mean: ['0.250', '0.250', '0.249', '0.251']
attL_top10_sum : ['0.250', '0.246', '0.244', '0.260']
attL_top25_mean: ['0.250', '0.250', '0.250', '0.250']
attL_top25_sum : ['0.253', '0.246', '0.242', '0.259']
attL_top50_mean: ['0.250', '0.250', '0.250', '0.250']
attL_top50_sum : ['0.252', '0.251', '0.250', '0.247']
===== 4. Chosen action and reward
Action: down
Reward: 0|Score: 5

─────── ===== Step 10 ===== ───────
===== 1. Textual obs:
o_desc : Forest Path This is a path winding through a dimly lit
    forest. The path heads north−south here. One particularly
    large tree with some low branches stands at the edge of the
    path.
o_inv : You are carrying: A jewel−encrusted egg You hear in the
    distance the chirping of a song bird.
o_feed : Forest Path
a_past : down
===== 2. Newly extracted triplets
[('Forest Path', 'has', 'exit to north'), ('Forest Path', 'has', '
    exit to south'), ('about 10 feet', 'down of', 'Forest Path'),
    ('all', 'in', 'Forest Path'), ('egg', 'in', 'Forest Path'), ('
    forest', 'in', 'Forest Path'), ('one large tree', 'is with', '
    low branches'), ('path', 'in', 'Forest Path'), ('path', '
    winding through', 'dimly lit forest'), ('path', 'winding
    through', 'forest'), ('path', 'winding through', 'lit forest')
    , ('tree', 'in', 'Forest Path'), ('you', 'have', ''), ('you',
    'have', 'hear in the distance the chirping of a song bird.'),
    ('you', 'have', 'jewel−encrusted egg')]
===== 3. Attention values:
─────── attH: o_desc, o_inv, o_feed, a_past
attH_max        : ['0.254', '0.253', '0.255', '0.238']
attH_mean       : ['0.322', '0.224', '0.248', '0.206']
```

```
attH_sum        : ['1.000', '0.000', '0.000', '0.000']
attH_top10_mean: ['0.297', '0.222', '0.296', '0.185']
attH_top10_sum : ['0.493', '0.027', '0.475', '0.004']
attH_top25_mean: ['0.337', '0.215', '0.275', '0.173']
attH_top25_sum : ['0.994', '0.000', '0.006', '0.000']
attH_top50_mean: ['0.356', '0.212', '0.252', '0.181']
attH_top50_sum : ['1.000', '0.000', '0.000', '0.000']
———— attL: connectivity, item_in_room, item_in_inv, history
attL_max        : ['0.250', '0.250', '0.249', '0.251']
attL_mean       : ['0.250', '0.250', '0.250', '0.250']
attL_sum        : ['0.251', '0.250', '0.250', '0.249']
attL_top10_mean: ['0.250', '0.250', '0.250', '0.251']
attL_top10_sum : ['0.250', '0.247', '0.247', '0.255']
attL_top25_mean: ['0.250', '0.250', '0.250', '0.250']
attL_top25_sum : ['0.252', '0.247', '0.246', '0.254']
attL_top50_mean: ['0.250', '0.250', '0.250', '0.250']
attL_top50_sum : ['0.251', '0.250', '0.250', '0.249']
===== 4. Chosen action and reward
Action: blow out trees
Reward: 0|Score: 5

———— ===== Step 11 ===== ————
===== 1. Textual obs:
o_desc: Forest Path This is a path winding through a dimly lit
    forest. The path heads north-south here. One particularly
    large tree with some low branches stands at the edge of the
    path. You hear in the distance the chirping of a song bird.
o_inv: You are carrying: A jewel-encrusted egg You hear in the
    distance the chirping of a song bird.
o_feed: You can't turn that off.
a_past: blow out trees
===== 2. Newly extracted triplets
[('Forest Path', 'has', 'exit to north'), ('Forest Path', 'has', '
    exit to south'), ('all', 'in', 'Forest Path'), ('egg', 'in', '
    Forest Path'), ('forest', 'in', 'Forest Path'), ('one large
    tree', 'is with', 'low branches'), ('path', 'in', 'Forest Path
    '), ('path', 'winding through', 'dimly lit forest'), ('path',
    'winding through', 'forest'), ('path', 'winding through', 'lit
    forest'), ('tree', 'in', 'Forest Path'), ('you', 'have', ''),
    ('you', 'have', 'hear in the distance the chirping of a song
    bird.'), ('you', 'have', 'jewel-encrusted egg'), ('you', 'hear
    ', 'chirping'), ('you', 'hear', 'chirping of song bird'), ('
    you', 'hear chirping in', 'distance')]
===== 3. Attention values:
———— attH: o_desc, o_inv, o_feed, a_past
attH_max        : ['0.254', '0.248', '0.254', '0.244']
attH_mean       : ['0.325', '0.215', '0.253', '0.207']
attH_sum        : ['1.000', '0.000', '0.000', '0.000']
attH_top10_mean: ['0.300', '0.203', '0.300', '0.197']
attH_top10_sum : ['0.491', '0.010', '0.492', '0.007']
attH_top25_mean: ['0.341', '0.192', '0.292', '0.175']
attH_top25_sum : ['0.980', '0.000', '0.020', '0.000']
attH_top50_mean: ['0.361', '0.195', '0.263', '0.181']
attH_top50_sum : ['1.000', '0.000', '0.000', '0.000']
———— attL: connectivity, item_in_room, item_in_inv, history
attL_max        : ['0.250', '0.249', '0.249', '0.252']
attL_mean       : ['0.250', '0.250', '0.250', '0.250']
attL_sum        : ['0.252', '0.250', '0.250', '0.247']
attL_top10_mean: ['0.250', '0.250', '0.250', '0.251']
```

```
attL_top10_sum : ['0.250', '0.246', '0.245', '0.258']
attL_top25_mean: ['0.250', '0.250', '0.250', '0.250']
attL_top25_sum : ['0.253', '0.246', '0.244', '0.257']
attL_top50_mean: ['0.250', '0.250', '0.250', '0.250']
attL_top50_sum : ['0.252', '0.250', '0.250', '0.247']
===== 4. Chosen action and reward
Action: south
Reward: 0|Score: 5

——— ===== Step 12 ===== ———
===== 1. Textual obs:
o_desc: North of House You are facing the north side of a white
    house. There is no door here, and all the windows are boarded
    up. To the north a narrow path winds through the trees.
o_inv: You are carrying: A jewel−encrusted egg
o_feed: North of House You are facing the north side of a white
    house. There is no door here, and all the windows are boarded
    up. To the north a narrow path winds through the trees.
a_past: south
===== 2. Newly extracted triplets
[('North of House', 'has', 'exit to north'), ('North of House', '
    has', 'exit to up'), ('about 10 feet', 'south of', 'north side
     of white house'), ('all', 'in', 'north side of white house'),
     ('egg', 'in', 'north side of white house'), ('house', 'in', '
    north side of white house'), ('narrow', 'in', 'north side of
    white house'), ('windows', 'in', 'north side of white house'),
     ('you', 'have', ''), ('you', 'have', 'jewel−encrusted egg'),
    ('you', 'in', 'north side'), ('you', 'in', 'north side of
    house'), ('you', 'in', 'north side of white house'), ('you', '
    in', 'side'), ('you', 'in', 'side of house'), ('you', 'in', '
    side of white house')]
===== 3. Attention values:
——— attH: o_desc, o_inv, o_feed, a_past
attH_max        : ['0.249', '0.254', '0.257', '0.241']
attH_mean       : ['0.265', '0.209', '0.275', '0.250']
attH_sum        : ['0.028', '0.000', '0.972', '0.000']
attH_top10_mean: ['0.267', '0.196', '0.287', '0.249']
attH_top10_sum : ['0.278', '0.013', '0.571', '0.138']
attH_top25_mean: ['0.271', '0.173', '0.307', '0.249']
attH_top25_sum : ['0.041', '0.000', '0.954', '0.005']
attH_top50_mean: ['0.273', '0.183', '0.296', '0.248']
attH_top50_sum : ['0.017', '0.000', '0.983', '0.000']
——— attL: connectivity, item_in_room, item_in_inv, history
attL_max        : ['0.250', '0.250', '0.250', '0.250']
attL_mean       : ['0.250', '0.250', '0.250', '0.250']
attL_sum         : ['0.252', '0.250', '0.250', '0.248']
attL_top10_mean: ['0.250', '0.250', '0.250', '0.250']
attL_top10_sum : ['0.250', '0.250', '0.249', '0.252']
attL_top25_mean: ['0.250', '0.250', '0.250', '0.250']
attL_top25_sum : ['0.251', '0.250', '0.248', '0.252']
attL_top50_mean: ['0.250', '0.250', '0.250', '0.250']
attL_top50_sum : ['0.252', '0.250', '0.250', '0.248']
===== 4. Chosen action and reward
Action: east
Reward: 0|Score: 5

——— ===== Step 13 ===== ———
===== 1. Textual obs:
```

```
o_desc: Behind House You are behind the white house. A path leads
    into the forest to the east. In one corner of the house there
    is a small window which is slightly ajar.
o_inv: You are carrying: A jewel−encrusted egg
o_feed: Behind House You are behind the white house. A path leads
    into the forest to the east. In one corner of the house there
    is a small window which is slightly ajar.
a_past: east
===== 2. Newly extracted triplets
[('Behind House', 'has', 'exit to east'), ('all', 'in', 'behind
    house white house'), ('egg', 'in', 'behind house white house')
    , ('house', 'in', 'behind house white house'), ('north side of
    white house', 'east of', 'behind house white house'), ('path'
    , 'in', 'behind house white house'), ('window', 'in', 'behind
    house white house'), ('you', 'have', ''), ('you', 'have', '
    jewel−encrusted egg'), ('you', 'in', 'behind house house'), ('
    you', 'in', 'behind house white house'), ('you', 'in', 'house'
    ), ('you', 'in', 'white house')]
===== 3. Attention values:
−−−−−− attH: o_desc, o_inv, o_feed, a_past
attH_max        : ['0.261', '0.248', '0.260', '0.231']
attH_mean       : ['0.288', '0.209', '0.277', '0.226']
attH_sum        : ['0.984', '0.000', '0.016', '0.000']
attH_top10_mean: ['0.292', '0.188', '0.291', '0.229']
attH_top10_sum : ['0.483', '0.006', '0.469', '0.042']
attH_top25_mean: ['0.306', '0.172', '0.305', '0.217']
attH_top25_sum : ['0.523', '0.000', '0.477', '0.000']
attH_top50_mean: ['0.309', '0.182', '0.297', '0.212']
attH_top50_sum : ['0.874', '0.000', '0.126', '0.000']
−−−−−− attL: connectivity, item_in_room, item_in_inv, history
attL_max        : ['0.250', '0.250', '0.250', '0.251']
attL_mean       : ['0.250', '0.250', '0.250', '0.250']
attL_sum        : ['0.252', '0.249', '0.251', '0.249']
attL_top10_mean: ['0.250', '0.250', '0.250', '0.250']
attL_top10_sum : ['0.250', '0.249', '0.248', '0.252']
attL_top25_mean: ['0.250', '0.250', '0.250', '0.250']
attL_top25_sum : ['0.251', '0.248', '0.248', '0.253']
attL_top50_mean: ['0.250', '0.250', '0.250', '0.250']
attL_top50_sum : ['0.252', '0.249', '0.251', '0.249']
===== 4. Chosen action and reward
Action: open window
Reward: 0|Score: 5

−−−−−− ===== Step 14 ===== −−−−−−
===== 1. Textual obs:
o_desc: Behind House You are behind the white house. A path leads
    into the forest to the east. In one corner of the house there
    is a small window which is open.
o_inv: You are carrying: A jewel−encrusted egg
o_feed: With great effort, you open the window far enough to allow
     entry.
a_past: open window
===== 2. Newly extracted triplets
[('Behind House', 'has', 'exit to east'), ('all', 'in', 'white
    house'), ('egg', 'in', 'white house'), ('house', 'in', 'white
    house'), ('path', 'in', 'white house'), ('small', 'in', 'white
     house'), ('you', 'have', ''), ('you', 'have', 'jewel−
    encrusted egg'), ('you', 'in', 'house'), ('you', 'in', 'white
    house')]
```

```
===== 3. Attention values:
——— attH: o_desc, o_inv, o_feed, a_past
attH_max        : ['0.276', '0.186', '0.277', '0.261']
attH_mean       : ['0.256', '0.207', '0.323', '0.214']
attH_sum        : ['0.000', '0.000', '1.000', '0.000']
attH_top10_mean: ['0.310', '0.176', '0.312', '0.201']
attH_top10_sum : ['0.478', '0.002', '0.514', '0.006']
attH_top25_mean: ['0.302', '0.167', '0.343', '0.187']
attH_top25_sum : ['0.038', '0.000', '0.962', '0.000']
attH_top50_mean: ['0.264', '0.174', '0.376', '0.186']
attH_top50_sum : ['0.000', '0.000', '1.000', '0.000']
——— attL: connectivity, item_in_room, item_in_inv, history
attL_max        : ['0.250', '0.250', '0.250', '0.250']
attL_mean       : ['0.250', '0.250', '0.250', '0.250']
attL_sum        : ['0.250', '0.250', '0.250', '0.250']
attL_top10_mean: ['0.250', '0.250', '0.250', '0.250']
attL_top10_sum : ['0.250', '0.250', '0.250', '0.250']
attL_top25_mean: ['0.250', '0.250', '0.250', '0.250']
attL_top25_sum : ['0.250', '0.250', '0.250', '0.250']
attL_top50_mean: ['0.250', '0.250', '0.250', '0.250']
attL_top50_sum : ['0.250', '0.250', '0.250', '0.250']
===== 4. Chosen action and reward
Action: west
Reward: 10|Score: 15

——— ===== Step 15 ===== ———
===== 1. Textual obs:
o_desc: Kitchen You are in the kitchen of the white house. A table
    seems to have been used recently for the preparation of food.
    A passage leads to the west and a dark staircase can be seen
    leading upward. A dark chimney leads down and to the east is a
    small window which is open. On the table is an elongated
    brown sack, smelling of hot peppers. A bottle is sitting on
    the table. The glass bottle contains: A quantity of water
o_inv: You are carrying: A jewel-encrusted egg
o_feed: Kitchen You are in the kitchen of the white house. A table
    seems to have been used recently for the preparation of food.
    A passage leads to the west and a dark staircase can be seen
    leading upward. A dark chimney leads down and to the east is a
    small window which is open. On the table is an elongated
    brown sack, smelling of hot peppers. A bottle is sitting on
    the table. The glass bottle contains: A quantity of water
a_past: west
===== 2. Newly extracted triplets
[('Kitchen', 'has', 'exit to down'), ('Kitchen', 'has', 'exit to
    east'), ('Kitchen', 'has', 'exit to up'), ('Kitchen', 'has', '
    exit to west'), ('all', 'in', 'kitchen of white house'), ('
    brown', 'in', 'kitchen of white house'), ('chimney', 'in', '
    kitchen of white house'), ('egg', 'in', 'kitchen of white
    house'), ('glass', 'in', 'kitchen of white house'), ('passage'
    , 'in', 'kitchen of white house'), ('staircase', 'in', '
    kitchen of white house'), ('table', 'in', 'kitchen of white
    house'), ('water', 'in', 'kitchen of white house'), ('white
    house', 'west of', 'kitchen of white house'), ('window', 'in',
     'kitchen of white house'), ('you', 'have', ''), ('you', 'have
    ', 'jewel-encrusted egg'), ('you', 'in', 'kitchen'), ('you', '
    in', 'kitchen of house'), ('you', 'in', 'kitchen of white
    house')]
===== 3. Attention values:
```

```
————  attH: o_desc, o_inv, o_feed, a_past
attH_max        : ['0.211', '0.218', '0.304', '0.267']
attH_mean       : ['0.246', '0.217', '0.275', '0.262']
attH_sum        : ['0.000', '0.000', '0.990', '0.010']
attH_top10_mean : ['0.208', '0.196', '0.338', '0.259']
attH_top10_sum  : ['0.007', '0.004', '0.925', '0.064']
attH_top25_mean : ['0.221', '0.198', '0.319', '0.263']
attH_top25_sum  : ['0.000', '0.000', '0.992', '0.008']
attH_top50_mean : ['0.235', '0.204', '0.297', '0.265']
attH_top50_sum  : ['0.000', '0.000', '0.997', '0.003']
————  attL: connectivity, item_in_room, item_in_inv, history
attL_max        : ['0.250', '0.250', '0.250', '0.250']
attL_mean       : ['0.250', '0.250', '0.250', '0.250']
attL_sum        : ['0.252', '0.250', '0.249', '0.248']
attL_top10_mean : ['0.250', '0.250', '0.250', '0.250']
attL_top10_sum  : ['0.251', '0.248', '0.248', '0.253']
attL_top25_mean : ['0.250', '0.250', '0.250', '0.250']
attL_top25_sum  : ['0.252', '0.248', '0.247', '0.253']
attL_top50_mean : ['0.250', '0.250', '0.250', '0.250']
attL_top50_sum  : ['0.252', '0.250', '0.249', '0.248']
===== 4. Chosen action and reward
Action: west
Reward: 0|Score: 15
```

## C.2   ztuu

```
————  ===== Step 1 ===== ————
===== 1. Textual obs:
o_desc: New Excavation You are standing at the entrance of a
    narrow, waist−high tunnel that winds out of sight to the
    northeast. This, you have been told, is the entrance to the
    newfound areas of the Great Underground Empire.
o_inv: You are carrying: a cheaply−made sword a brass lantern (
    providing light)
o_feed: It is with great hope and expectation I send you on this
    mission," the Grand Inquisitor intones, his voice booming like
     distant, rumbling thunderheads. "We have discovered what
    appears to be a piece of the Great Underground Empire, never
    before explored. I would have sent Grubald the Bold, but he's
    busy. As is Matchlick the Mighty. So, 'tis you." "What of
    Linklaw the Lucky?" you ask shyly. "Laid up. I'm afraid 'tis
    you and none other. A simple, solo quest, one of fun and mirth
    , mischief and merriment, a jaunt, an excursion, a simple
    outing. Merely explore, enjoy yourself, and bring back news of
     what lies there, long−ago buried for countless ages." "And
    Fretsham the Fortunate?" you ask hopefully. "A fatality!" the
    Inquisitor booms. "Here," he says with evaporating patience, "
    take these and be of good cheer." He thrusts into your hands
    two things: a brass lantern of dubious quality; and a plastic
    sword of no antiquity or use whatsoever. "But what of Kolchack
     the —" "Enough! Kolchack is contracted. Remember, who is the
     boss of you!" He pauses briefly as if awaiting a reply. Then
     without warning, he answers his own question. "Me! I am the
    boss of you! Now begone!" And with that, you embark on... Zork
    : The Undiscovered Underground, a text adventure prequel to
    Zork Grand Inquisitor Copyright 1997 by Activision Inc.
    Written by Marc Blank and Mike Berlyn, programmed by Gerry
    Kevin Wilson. Release 16 / Serial number 970828 / Inform v6.13
     Library 6/2 Standard interpreter 1.0 New Excavation You are
```

standing at the entrance of a narrow, waist−high tunnel that
winds out of sight to the northeast. This, you have been told,
 is the entrance to the newfound areas of the Great
Underground Empire. In your left hand hangs a rather
unconvincing replica of the traditional adventurer's brass
lantern. In the other is an ersatz−Elvish sword of no
antiquity whatsoever. Its plastic blade, dull as a bread knife
, barely reflects the lantern's dim and uneven glow.
a_past: look
===== 2. newly extracted triplets
[('New Excavation', 'has', 'exit to east'), ('New Excavation', '
    has', 'exit to north'), ('New Excavation', 'has', 'exit to
    northeast'), ('New Excavation', 'has', 'exit to up'), ('all',
    'in', 'entrance of narrow waist high tunnel'), ('lantern', 'in
    ', 'entrance of narrow waist high tunnel'), ('northeast', 'in
    ', 'entrance of narrow waist high tunnel'), ('other', 'in', '
    entrance of narrow waist high tunnel'), ('sword', 'in', '
    entrance of narrow waist high tunnel'), ('tunnel', 'in', '
    entrance of narrow waist high tunnel'), ('you', 'have', 'brass
     lantern (providing light)'), ('you', 'have', 'cheaply−made
    sword'), ('you', 'in', 'entrance'), ('you', 'in', 'entrance of
     narrow waist high tunnel'), ('you', 'in', 'entrance of narrow
     waist tunnel'), ('you', 'in', 'entrance of waist high tunnel
    '), ('you', 'in', 'entrance of waist tunnel')]
===== 3. Attention values:
——— attH: o_desc, o_inv, o_feed, a_past
attH_max         : ['0.223', '0.247', '0.257', '0.273']
attH_mean        : ['0.255', '0.240', '0.238', '0.268']
attH_sum         : ['0.006', '0.000', '0.000', '0.994']
attH_top10_mean: ['0.229', '0.235', '0.223', '0.313']
attH_top10_sum : ['0.039', '0.049', '0.029', '0.883']
attH_top25_mean: ['0.230', '0.223', '0.212', '0.335']
attH_top25_sum : ['0.000', '0.000', '0.000', '1.000']
attH_top50_mean: ['0.250', '0.230', '0.224', '0.296']
attH_top50_sum : ['0.000', '0.000', '0.000', '1.000']
——— attL: connectivity, item_in_room, item_in_inv, history
attL_max         : ['0.245', '0.255', '0.249', '0.251']
attL_mean        : ['0.251', '0.242', '0.253', '0.253']
attL_sum         : ['0.251', '0.042', '0.328', '0.379']
attL_top10_mean: ['0.247', '0.249', '0.250', '0.253']
attL_top10_sum : ['0.223', '0.243', '0.252', '0.282']
attL_top25_mean: ['0.249', '0.246', '0.251', '0.253']
attL_top25_sum : ['0.215', '0.165', '0.279', '0.341']
attL_top50_mean: ['0.251', '0.242', '0.253', '0.253']
attL_top50_sum : ['0.251', '0.042', '0.328', '0.379']
===== 4. Chosen action and reward
Action: turn brass off
Reward: 0|Score: 0

——— ===== Step 2 ===== ———
===== 1. Textual obs:
o_desc: New Excavation You are standing at the entrance of a
    narrow, waist−high tunnel that winds out of sight to the
    northeast. This, you have been told, is the entrance to the
    newfound areas of the Great Underground Empire.
o_inv: You are carrying: a cheaply−made sword a brass lantern
o_feed: The lantern is now off. You had better hope that it comes
    back on when you come to your senses.
a_past: turn brass off

```
===== 2. newly extracted triplets
[('all', 'in', 'entrance of narrow waist high tunnel'), ('lantern
    ', 'in', 'entrance of narrow waist high tunnel'), ('northeast
    ', 'in', 'entrance of narrow waist high tunnel'), ('sword', '
    in', 'entrance of narrow waist high tunnel'), ('tunnel', 'in',
     'entrance of narrow waist high tunnel'), ('you', 'have', '
    brass lantern'), ('you', 'have', 'cheaply-made sword'), ('you
    ', 'in', 'entrance'), ('you', 'in', 'entrance of narrow waist
    high tunnel'), ('you', 'in', 'entrance of narrow waist tunnel
    '), ('you', 'in', 'entrance of waist high tunnel'), ('you', '
    in', 'entrance of waist tunnel')]
===== 3. Attention values:
——— attH: o_desc, o_inv, o_feed, a_past
attH_max        : ['0.275', '0.242', '0.209', '0.275']
attH_mean       : ['0.321', '0.213', '0.204', '0.262']
attH_sum        : ['1.000', '0.000', '0.000', '0.000']
attH_top10_mean: ['0.312', '0.214', '0.161', '0.313']
attH_top10_sum : ['0.493', '0.011', '0.001', '0.495']
attH_top25_mean: ['0.339', '0.181', '0.153', '0.327']
attH_top25_sum : ['0.700', '0.000', '0.000', '0.300']
attH_top50_mean: ['0.374', '0.182', '0.167', '0.277']
attH_top50_sum : ['1.000', '0.000', '0.000', '0.000']
——— attL: connectivity, item_in_room, item_in_inv, history
attL_max        : ['0.248', '0.251', '0.251', '0.249']
attL_mean       : ['0.250', '0.248', '0.251', '0.250']
attL_sum        : ['0.253', '0.174', '0.310', '0.263']
attL_top10_mean: ['0.249', '0.250', '0.251', '0.249']
attL_top10_sum : ['0.240', '0.250', '0.265', '0.245']
attL_top25_mean: ['0.250', '0.249', '0.251', '0.250']
attL_top25_sum : ['0.237', '0.228', '0.289', '0.246']
attL_top50_mean: ['0.250', '0.248', '0.251', '0.250']
attL_top50_sum : ['0.253', '0.174', '0.310', '0.263']
===== 4. Chosen action and reward
Action: throw brass
Reward: 5|Score: 5

——— ===== Step 3 ===== ———
===== 1. Textual obs:
o_desc: New Excavation You are standing at the entrance of a
    narrow, waist-high tunnel that winds out of sight to the
    northeast. This, you have been told, is the entrance to the
    newfound areas of the Great Underground Empire.
o_inv: You are carrying: a cheaply-made sword a brass lantern (
    providing light)
o_feed: You throw the lamp, and quickly retrieve it. Your rough
    treatment of the lantern was enough to get it lit again. This
    must be your lucky day. [Your score has just gone up by five
    points.]
a_past: throw brass
===== 2. newly extracted triplets
[('New Excavation', 'has', 'exit to up'), ('all', 'in', 'entrance
    of narrow waist high tunnel'), ('lantern', 'in', 'entrance of
    narrow waist high tunnel'), ('northeast', 'in', 'entrance of
    narrow waist high tunnel'), ('sword', 'in', 'entrance of
    narrow waist high tunnel'), ('tunnel', 'in', 'entrance of
    narrow waist high tunnel'), ('you', 'have', 'brass lantern (
    providing light)'), ('you', 'have', 'cheaply-made sword'), ('
    you', 'in', 'entrance'), ('you', 'in', 'entrance of narrow
    waist high tunnel'), ('you', 'in', 'entrance of narrow waist
```

tunnel'), ('you', 'in', 'entrance of waist high tunnel'), ('
    you', 'in', 'entrance of waist tunnel')]
===== 3. Attention values:
———— attH: o_desc, o_inv, o_feed, a_past
attH_max        : ['0.250', '0.240', '0.255', '0.255']
attH_mean       : ['0.262', '0.227', '0.243', '0.268']
attH_sum        : ['0.111', '0.000', '0.000', '0.889']
attH_top10_mean: ['0.263', '0.215', '0.243', '0.279']
attH_top10_sum : ['0.295', '0.039', '0.131', '0.535']
attH_top25_mean: ['0.272', '0.199', '0.231', '0.298']
attH_top25_sum : ['0.094', '0.000', '0.002', '0.904']
attH_top50_mean: ['0.272', '0.206', '0.237', '0.285']
attH_top50_sum : ['0.087', '0.000', '0.000', '0.913']
———— attL: connectivity, item_in_room, item_in_inv, history
attL_max        : ['0.251', '0.249', '0.250', '0.250']
attL_mean       : ['0.250', '0.250', '0.250', '0.250']
attL_sum        : ['0.258', '0.244', '0.244', '0.254']
attL_top10_mean: ['0.250', '0.250', '0.250', '0.250']
attL_top10_sum : ['0.254', '0.247', '0.247', '0.252']
attL_top25_mean: ['0.250', '0.250', '0.250', '0.250']
attL_top25_sum : ['0.257', '0.244', '0.245', '0.254']
attL_top50_mean: ['0.250', '0.250', '0.250', '0.250']
attL_top50_sum : ['0.258', '0.244', '0.244', '0.254']
===== 4. Chosen action and reward
Action: turn lantern off
Reward: 0|Score: 5

———— ===== Step 4 ===== ————
===== 1. Textual obs:
o_desc: New Excavation You are standing at the entrance of a
    narrow, waist-high tunnel that winds out of sight to the
    northeast. This, you have been told, is the entrance to the
    newfound areas of the Great Underground Empire.
o_inv: You are carrying: a cheaply-made sword a brass lantern
o_feed: The lantern is now off. You had better hope that it comes
    back on when you come to your senses.
a_past: turn lantern off
===== 2. newly extracted triplets
[('all', 'in', 'entrance of narrow waist high tunnel'), ('lantern
    ', 'in', 'entrance of narrow waist high tunnel'), ('northeast
    ', 'in', 'entrance of narrow waist high tunnel'), ('sword', '
    in', 'entrance of narrow waist high tunnel'), ('tunnel', 'in',
    'entrance of narrow waist high tunnel'), ('you', 'have', '
    brass lantern'), ('you', 'have', 'cheaply-made sword'), ('you
    ', 'in', 'entrance'), ('you', 'in', 'entrance of narrow waist
    high tunnel'), ('you', 'in', 'entrance of narrow waist tunnel
    '), ('you', 'in', 'entrance of waist high tunnel'), ('you', '
    in', 'entrance of waist tunnel')]
===== 3. Attention values:
———— attH: o_desc, o_inv, o_feed, a_past
attH_max        : ['0.279', '0.212', '0.227', '0.282']
attH_mean       : ['0.269', '0.236', '0.224', '0.271']
attH_sum        : ['0.330', '0.000', '0.000', '0.670']
attH_top10_mean: ['0.294', '0.205', '0.191', '0.311']
attH_top10_sum : ['0.360', '0.010', '0.005', '0.625']
attH_top25_mean: ['0.294', '0.204', '0.185', '0.318']
attH_top25_sum : ['0.117', '0.000', '0.000', '0.883']
attH_top50_mean: ['0.285', '0.218', '0.201', '0.295']
attH_top50_sum : ['0.147', '0.000', '0.000', '0.853']

```
————— attL : connectivity , item_in_room , item_in_inv , history
attL_max      : ['0.249', '0.252', '0.250', '0.248']
attL_mean     : ['0.249', '0.250', '0.250', '0.250']
attL_sum      : ['0.219', '0.274', '0.265', '0.241']
attL_top10_mean: ['0.249', '0.252', '0.251', '0.249']
attL_top10_sum : ['0.240', '0.266', '0.256', '0.238']
attL_top25_mean: ['0.249', '0.251', '0.251', '0.249']
attL_top25_sum : ['0.232', '0.275', '0.262', '0.231']
attL_top50_mean: ['0.249', '0.250', '0.250', '0.250']
attL_top50_sum : ['0.219', '0.274', '0.265', '0.241']
===== 4. Chosen action and reward
Action: take into sword
Reward: 0|Score: 5

————— ===== Step 5 ===== —————
===== 1. Textual obs :
o_desc: New Excavation You are standing at the entrance of a
   narrow , waist−high tunnel that winds out of sight to the
   northeast. This , you have been told , is the entrance to the
   newfound areas of the Great Underground Empire .
o_inv: You are carrying: a cheaply−made sword a brass lantern
o_feed: That's not something you can enter .
a_past: take into sword
===== 2. newly extracted triplets
[('all', 'in', 'entrance of narrow waist high tunnel'), ('lantern
   ', 'in', 'entrance of narrow waist high tunnel'), ('northeast
   ', 'in', 'entrance of narrow waist high tunnel'), ('sword', '
   in', 'entrance of narrow waist high tunnel'), ('tunnel', 'in',
    'entrance of narrow waist high tunnel'), ('you', 'have', '
   brass lantern'), ('you', 'have', 'cheaply−made sword'), ('you
   ', 'in', 'entrance'), ('you', 'in', 'entrance of narrow waist
   high tunnel'), ('you', 'in', 'entrance of narrow waist tunnel
   '), ('you', 'in', 'entrance of waist high tunnel'), ('you', '
   in', 'entrance of waist tunnel')]
===== 3. Attention values :
————— attH : o_desc , o_inv , o_feed , a_past
attH_max      : ['0.272', '0.205', '0.248', '0.275']
attH_mean     : ['0.265', '0.233', '0.225', '0.277']
attH_sum      : ['0.012', '0.000', '0.000', '0.988']
attH_top10_mean: ['0.288', '0.203', '0.202', '0.306']
attH_top10_sum : ['0.344', '0.011', '0.010', '0.635']
attH_top25_mean: ['0.283', '0.202', '0.191', '0.324']
attH_top25_sum : ['0.032', '0.000', '0.000', '0.968']
attH_top50_mean: ['0.276', '0.216', '0.205', '0.303']
attH_top50_sum : ['0.009', '0.000', '0.000', '0.991']
————— attL : connectivity , item_in_room , item_in_inv , history
attL_max      : ['0.249', '0.252', '0.251', '0.248']
attL_mean     : ['0.249', '0.250', '0.251', '0.250']
attL_sum      : ['0.214', '0.269', '0.276', '0.241']
attL_top10_mean: ['0.249', '0.252', '0.251', '0.248']
attL_top10_sum : ['0.238', '0.265', '0.262', '0.235']
attL_top25_mean: ['0.249', '0.251', '0.251', '0.249']
attL_top25_sum : ['0.227', '0.275', '0.272', '0.226']
attL_top50_mean: ['0.249', '0.250', '0.251', '0.250']
attL_top50_sum : ['0.214', '0.269', '0.276', '0.241']
===== 4. Chosen action and reward
Action: throw brass
Reward: 0|Score: 5
```

―――― ===== Step 6 ===== ――――
===== 1. Textual obs:
o_desc: New Excavation You are standing at the entrance of a
    narrow, waist-high tunnel that winds out of sight to the
    northeast. This, you have been told, is the entrance to the
    newfound areas of the Great Underground Empire.
o_inv: You are carrying: a cheaply-made sword a brass lantern (
    providing light)
o_feed: You throw the lamp, and quickly retrieve it. Your rough
    treatment of the lantern was enough to get it lit again. This
    must be your lucky day.
a_past: throw brass
===== 2. newly extracted triplets
[('all', 'in', 'entrance of narrow waist high tunnel'), ('lantern
    ', 'in', 'entrance of narrow waist high tunnel'), ('northeast
    ', 'in', 'entrance of narrow waist high tunnel'), ('sword', '
    in', 'entrance of narrow waist high tunnel'), ('tunnel', 'in',
     'entrance of narrow waist high tunnel'), ('you', 'have', '
    brass lantern (providing light)'), ('you', 'have', 'cheaply-
    made sword'), ('you', 'in', 'entrance'), ('you', 'in', '
    entrance of narrow waist high tunnel'), ('you', 'in', '
    entrance of narrow waist tunnel'), ('you', 'in', 'entrance of
    waist high tunnel'), ('you', 'in', 'entrance of waist tunnel')
    ]
===== 3. Attention values:
―――― attH: o_desc, o_inv, o_feed, a_past
attH_max       : ['0.249', '0.242', '0.253', '0.255']
attH_mean      : ['0.268', '0.228', '0.230', '0.274']
attH_sum       : ['0.104', '0.000', '0.000', '0.896']
attH_top10_mean: ['0.272', '0.220', '0.218', '0.290']
attH_top10_sum : ['0.320', '0.039', '0.034', '0.607']
attH_top25_mean: ['0.283', '0.205', '0.203', '0.308']
attH_top25_sum : ['0.104', '0.000', '0.000', '0.896']
attH_top50_mean: ['0.283', '0.209', '0.212', '0.297']
attH_top50_sum : ['0.083', '0.000', '0.000', '0.917']
―――― attL: connectivity, item_in_room, item_in_inv, history
attL_max       : ['0.250', '0.250', '0.250', '0.250']
attL_mean      : ['0.250', '0.250', '0.250', '0.250']
attL_sum       : ['0.251', '0.244', '0.250', '0.255']
attL_top10_mean: ['0.250', '0.250', '0.250', '0.250']
attL_top10_sum : ['0.251', '0.249', '0.249', '0.251']
attL_top25_mean: ['0.250', '0.250', '0.250', '0.250']
attL_top25_sum : ['0.252', '0.247', '0.249', '0.253']
attL_top50_mean: ['0.250', '0.250', '0.250', '0.250']
attL_top50_sum : ['0.251', '0.244', '0.250', '0.255']
===== 4. Chosen action and reward
Action: take into other
Reward: 0|Score: 5

―――― ===== Step 7 ===== ――――
===== 1. Textual obs:
o_desc: Tunnel You are in a narrow tunnel whose entrance has been
    blocked by gigantic, translucent boulders. Your only option is
     to continue to the northeast, where the tunnel gradually
    widens.
o_inv: You are carrying: a cheaply-made sword a brass lantern (
    providing light)
o_feed: (the tunnel) As you enter the tunnel, you feel a slight
    rumble beneath your crepe-soled boots. As you turn to look

back up to the exit, giant, translucent boulders begin to fall
from the walls and ceiling. You duck and cover your head —
when the dust clears, you feel your heart sink in your chest.
You are trapped! Tunnel You are in a narrow tunnel whose
entrance has been blocked by gigantic, translucent boulders.
Your only option is to continue to the northeast, where the
tunnel gradually widens.
a_past: take into other
===== 2. newly extracted triplets
[('Tunnel', 'has', 'exit to east'), ('Tunnel', 'has', 'exit to
north'), ('Tunnel', 'has', 'exit to northeast'), ('Tunnel', '
has', 'exit to up'), ('all', 'in', 'narrow tunnel'), ('
boulders', 'in', 'narrow tunnel'), ('ceiling', 'in', 'narrow
tunnel'), ('entrance', 'has', 'has blocked by gigantic
translucent boulders'), ('lantern', 'in', 'narrow tunnel'), ('
northeast', 'in', 'narrow tunnel'), ('sword', 'in', 'narrow
tunnel'), ('tunnel', 'in', 'narrow tunnel'), ('you', 'have', '
brass lantern (providing light)'), ('you', 'have', 'cheaply−
made sword'), ('you', 'in', 'narrow'), ('you', 'in', 'narrow
tunnel'), ('you', 'in', 'tunnel')]
===== 3. Attention values:
───── attH: o_desc, o_inv, o_feed, a_past
attH_max        : ['0.251', '0.253', '0.243', '0.252']
attH_mean       : ['0.277', '0.245', '0.228', '0.249']
attH_sum        : ['1.000', '0.000', '0.000', '0.000']
attH_top10_mean: ['0.265', '0.262', '0.205', '0.268']
attH_top10_sum : ['0.327', '0.288', '0.024', '0.361']
attH_top25_mean: ['0.293', '0.252', '0.190', '0.265']
attH_top25_sum : ['0.903', '0.021', '0.000', '0.076']
attH_top50_mean: ['0.303', '0.241', '0.207', '0.249']
attH_top50_sum : ['1.000', '0.000', '0.000', '0.000']
───── attL: connectivity, item_in_room, item_in_inv, history
attL_max        : ['0.250', '0.251', '0.250', '0.250']
attL_mean       : ['0.250', '0.250', '0.250', '0.250']
attL_sum        : ['0.252', '0.247', '0.249', '0.251']
attL_top10_mean: ['0.250', '0.250', '0.250', '0.250']
attL_top10_sum : ['0.248', '0.253', '0.249', '0.249']
attL_top25_mean: ['0.250', '0.250', '0.250', '0.250']
attL_top25_sum : ['0.248', '0.255', '0.248', '0.249']
attL_top50_mean: ['0.250', '0.250', '0.250', '0.250']
attL_top50_sum : ['0.252', '0.247', '0.249', '0.251']
===== 4. Chosen action and reward
Action: east
Reward: 0|Score: 5

─────  ===== Step 8 ===== ─────
===== 1. Textual obs:
o_desc: Cultural Complex This imposing ante−room, the center of
what was apparently the cultural center of the GUE, is adorned
in the ghastly style of the GUE's "Grotesque Period." With
leering gargoyles, cartoonish friezes depicting long−forgotten
scenes of GUE history, and primitive statuary of pointy−
headed personages unknown (perhaps very, very distant
progenitors of the Flatheads), the place would have been best
left undiscovered. North of here, a large hallway passes under
the roughly hewn inscription "Convention Center." To the
east, under a fifty−story triumphal arch, a passageway the
size of a large city boulevard opens into the Royal Theater. A
relatively small and unobtrusive sign (perhaps ten feet high)

stands nearby. South, a smaller and more dignified (i.e. post
    -Dimwit) path leads into what is billed as the "Hall of
    Science."
o_inv: You are carrying: a cheaply-made sword a brass lantern (
    providing light)
o_feed: Cultural Complex This imposing ante-room, the center of
    what was apparently the cultural center of the GUE, is adorned
    in the ghastly style of the GUE's "Grotesque Period." With
    leering gargoyles, cartoonish friezes depicting long-forgotten
    scenes of GUE history, and primitive statuary of pointy-
    headed personages unknown (perhaps very, very distant
    progenitors of the Flatheads), the place would have been best
    left undiscovered. North of here, a large hallway passes under
    the roughly hewn inscription "Convention Center." To the
    east, under a fifty-story triumphal arch, a passageway the
    size of a large city boulevard opens into the Royal Theater. A
    relatively small and unobtrusive sign (perhaps ten feet high)
    stands nearby. South, a smaller and more dignified (i.e. post
    -Dimwit) path leads into what is billed as the "Hall of
    Science."
a_past: east
===== 2. newly extracted triplets
[('Cultural Complex', 'has', 'exit to east'), ('all', 'in', '
    Cultural Complex'), ('ante room', 'is', 'adorned'), ('ante
    room', 'is adorned in', 'ghastly style'), ('ante room', 'is
    adorned in', "ghastly style of gue 's grotesque period"), ('
    ante room', 'is adorned in', 'style'), ('ante room', 'is
    adorned in', "style of gue 's grotesque period"), ('cartoonish
    friezes', 'headed', 'personages'), ('cartoonish friezes', '
    headed', 'personages unknown'), ('depicting', 'forgotten
    scenes of', 'gue history'), ('depicting', 'long forgotten
    scenes of', 'gue history'), ('depicting', 'long scenes of', '
    gue history'), ('depicting', 'scenes of', 'gue history'), ('
    east', 'in', 'Cultural Complex'), ('friezes', 'headed', '
    personages'), ('friezes', 'headed', 'personages unknown'), ('
    gue', 'of', 'grotesque period'), ('hallway', 'passes under', '
    hewn inscription'), ('hallway', 'passes under', 'inscription')
    , ('hallway', 'passes under', 'roughly hewn inscription'), ('
    imposing ante room', 'is', 'adorned'), ('imposing ante room',
    'is adorned in', 'ghastly style'), ('imposing ante room', 'is
    adorned in', "ghastly style of gue 's grotesque period"), ('
    imposing ante room', 'is adorned in', 'style'), ('imposing
    ante room', 'is adorned in', "style of gue 's grotesque period
    "), ('lantern', 'in', 'Cultural Complex'), ('large hallway', '
    passes under', 'hewn inscription'), ('large hallway', 'passes
    under', 'inscription'), ('large hallway', 'passes under', '
    roughly hewn inscription'), ('narrow tunnel', 'east of', '
    Cultural Complex'), ('primitive statuary', 'headed', '
    personages'), ('primitive statuary', 'headed', 'personages
    unknown'), ('primitive statuary', 'leering', 'gargoyles'), ('
    sign', 'in', 'Cultural Complex'), ('statuary', 'headed', '
    personages'), ('statuary', 'headed', 'personages unknown'), ('
    statuary', 'in', 'Cultural Complex'), ('statuary', 'leering',
    'gargoyles'), ('sword', 'in', 'Cultural Complex'), ('you', '
    have', 'brass lantern (providing light)'), ('you', 'have', '
    cheaply-made sword')]
===== 3. Attention values:
------- attH: o_desc, o_inv, o_feed, a_past
attH_max       : ['0.251', '0.251', '0.248', '0.251']

```
attH_mean      : ['0.282', '0.244', '0.231', '0.242']
attH_sum       : ['1.000', '0.000', '0.000', '0.000']
attH_top10_mean: ['0.262', '0.257', '0.224', '0.257']
attH_top10_sum : ['0.353', '0.288', '0.074', '0.285']
attH_top25_mean: ['0.302', '0.249', '0.204', '0.245']
attH_top25_sum : ['0.987', '0.008', '0.000', '0.005']
attH_top50_mean: ['0.314', '0.238', '0.213', '0.234']
attH_top50_sum : ['1.000', '0.000', '0.000', '0.000']
————— attL: connectivity, item_in_room, item_in_inv, history
attL_max       : ['0.249', '0.254', '0.248', '0.249']
attL_mean      : ['0.251', '0.250', '0.250', '0.250']
attL_sum       : ['0.284', '0.242', '0.226', '0.248']
attL_top10_mean: ['0.249', '0.252', '0.249', '0.250']
attL_top10_sum : ['0.243', '0.266', '0.241', '0.250']
attL_top25_mean: ['0.250', '0.251', '0.249', '0.250']
attL_top25_sum : ['0.247', '0.268', '0.234', '0.251']
attL_top50_mean: ['0.251', '0.250', '0.250', '0.250']
attL_top50_sum : ['0.284', '0.242', '0.226', '0.248']
===== 4. Chosen action and reward
Action: north
Reward: 0|Score: 5

————— ===== Step 9 ===== —————
===== 1. Textual obs:
o_desc: Convention Center Lobby You are standing at the entrance
    of the GUE Convention Center. On the floor, you can see
    occasional splotches of paint through the grime and dust.
    Although the whole cultural center appears to have been long−
    abandoned, loud, low−pitched sounds emanate from the door
    standing open to the north. Next to the door is a sign, and to
     the sign's left (northwest to you) is a souvenir stand. The
    cultural center's lobby lies to the south.
o_inv: You are carrying: a cheaply−made sword a brass lantern (
    providing light)
o_feed: Convention Center Lobby You are standing at the entrance
    of the GUE Convention Center. On the floor, you can see
    occasional splotches of paint through the grime and dust.
    Although the whole cultural center appears to have been long−
    abandoned, loud, low−pitched sounds emanate from the door
    standing open to the north. And while these sounds most
    closely approximate those made by under−lubricated granola
    fabricators, they nonetheless fill you with dread. Next to the
     door is a sign, and to the sign's left (northwest to you) is
    a souvenir stand. The cultural center's lobby lies to the
    south.
a_past: north
===== 2. newly extracted triplets
[('Convention Center Lobby', 'has', 'exit to north'), ('Convention
     Center Lobby', 'has', 'exit to northwest'), ('Convention
    Center Lobby', 'has', 'exit to south'), ('Convention Center
    Lobby', 'has', 'exit to west'), ('all', 'in', 'entrance of gue
     convention center'), ('door', 'in', 'entrance of gue
    convention center'), ('floor', 'in', 'entrance of gue
    convention center'), ('lantern', 'in', 'entrance of gue
    convention center'), ('narrow tunnel', 'north of', 'entrance
    of gue convention center'), ('north', 'in', 'entrance of gue
    convention center'), ('paint', 'in', 'entrance of gue
    convention center'), ('sign', 'in', 'entrance of gue
    convention center'), ('south', 'in', 'entrance of gue
```

```
convention center'), ('sword', 'in', 'entrance of gue
    convention center'), ('you', 'have', 'brass lantern (providing
    light)'), ('you', 'have', 'cheaply-made sword'), ('you', 'in
    ', 'entrance'), ('you', 'in', 'entrance of gue convention
    center')]
===== 3. Attention values:
——— attH: o_desc, o_inv, o_feed, a_past
attH_max        : ['0.251', '0.251', '0.248', '0.250']
attH_mean       : ['0.297', '0.240', '0.225', '0.237']
attH_sum        : ['1.000', '0.000', '0.000', '0.000']
attH_top10_mean : ['0.266', '0.254', '0.220', '0.261']
attH_top10_sum  : ['0.380', '0.240', '0.057', '0.322']
attH_top25_mean : ['0.313', '0.247', '0.197', '0.243']
attH_top25_sum  : ['0.996', '0.003', '0.000', '0.002']
attH_top50_mean : ['0.347', '0.229', '0.201', '0.223']
attH_top50_sum  : ['1.000', '0.000', '0.000', '0.000']
——— attL: connectivity, item_in_room, item_in_inv, history
attL_max        : ['0.247', '0.249', '0.249', '0.254']
attL_mean       : ['0.250', '0.250', '0.250', '0.250']
attL_sum        : ['0.257', '0.250', '0.243', '0.250']
attL_top10_mean : ['0.249', '0.250', '0.250', '0.251']
attL_top10_sum  : ['0.238', '0.254', '0.248', '0.260']
attL_top25_mean : ['0.249', '0.250', '0.250', '0.250']
attL_top25_sum  : ['0.237', '0.260', '0.246', '0.258']
attL_top50_mean : ['0.250', '0.250', '0.250', '0.250']
attL_top50_sum  : ['0.257', '0.250', '0.243', '0.250']
===== 4. Chosen action and reward
Action: northwest
Reward: 0|Score: 5

——— ===== Step 10 ===== ———
===== 1. Textual obs:
o_desc: Souvenir Stand You find yourself standing in front of a
    souvenir stand that is clearly affiliated with the Convention
    Center. An odd-looking salesman sits behind the counter, his
    head resting on his hand and his eyes glued to a spot
    somewhere above and behind you. A notice on the counter points
    toward a small, curtained room to the north. On the counter
    are four candy bars (a Baby Rune, a Multi-Implementeers, a
    ZM$100000 and a Forever Gores), some popcorn, some postcards
    and a chicken finger. You can see a salesman here.
o_inv: You are carrying: a cheaply-made sword a brass lantern (
    providing light)
o_feed: Souvenir Stand You find yourself standing in front of a
    souvenir stand that is clearly affiliated with the Convention
    Center. An odd-looking salesman sits behind the counter, his
    head resting on his hand and his eyes glued to a spot
    somewhere above and behind you (a quick, frightened glance
    over your shoulder reveals nothing eye-catching; indeed,
    nothing at all.) A notice on the counter points toward a small
    , curtained room to the north. On the counter are four candy
    bars (a Baby Rune, a Multi-Implementeers, a ZM$100000 and a
    Forever Gores), some popcorn, some postcards and a chicken
    finger. You can see a salesman here.
a_past: northwest
===== 2. newly extracted triplets
[('Souvenir Stand', 'has', 'exit to north'), ('all', 'in', '
    Souvenir Stand'), ('candy', 'in', 'Souvenir Stand'), ('counter
    ', 'in', 'Souvenir Stand'), ('entrance of gue convention
```

center', 'north of', 'Souvenir Stand'), ('finger', 'in', '
Souvenir Stand'), ('gores', 'in', 'Souvenir Stand'), ('his
eyes', 'glued to', 'spot'), ('his eyes', 'glued to', 'spot
above'), ('his eyes', 'glued to', 'spot somewhere above'), ('
his head', 'resting on', 'his hand'), ('lantern', 'in', '
Souvenir Stand'), ('looking salesman', 'sits behind', 'counter
'), ('multi', 'in', 'Souvenir Stand'), ('north', 'in', '
Souvenir Stand'), ('notice', 'in', 'Souvenir Stand'), ('odd
looking salesman', 'sits behind', 'counter'), ('odd salesman',
'sits behind', 'counter'), ('popcorn', 'in', 'Souvenir Stand
'), ('postcards', 'in', 'Souvenir Stand'), ('rune', 'in', '
Souvenir Stand'), ('salesman', 'in', 'Souvenir Stand'), ('
salesman', 'sits behind', 'counter'), ('sword', 'in', '
Souvenir Stand'), ('you', 'can see', 'salesman'), ('you', 'can
see here', 'salesman'), ('you', 'have', 'brass lantern (
providing light)'), ('you', 'have', 'cheaply−made sword')]
===== 3. Attention values:
——— attH: o_desc, o_inv, o_feed, a_past
attH_max        : ['0.250', '0.250', '0.249', '0.250']
attH_mean       : ['0.304', '0.235', '0.228', '0.233']
attH_sum        : ['1.000', '0.000', '0.000', '0.000']
attH_top10_mean: ['0.261', '0.247', '0.237', '0.255']
attH_top10_sum : ['0.367', '0.205', '0.138', '0.289']
attH_top25_mean: ['0.317', '0.238', '0.207', '0.238']
attH_top25_sum : ['0.998', '0.001', '0.000', '0.001']
attH_top50_mean: ['0.362', '0.218', '0.205', '0.215']
attH_top50_sum : ['1.000', '0.000', '0.000', '0.000']
——— attL: connectivity, item_in_room, item_in_inv, history
attL_max        : ['0.250', '0.251', '0.249', '0.250']
attL_mean       : ['0.250', '0.250', '0.250', '0.250']
attL_sum        : ['0.253', '0.266', '0.241', '0.241']
attL_top10_mean: ['0.250', '0.251', '0.249', '0.250']
attL_top10_sum : ['0.246', '0.260', '0.244', '0.250']
attL_top25_mean: ['0.250', '0.251', '0.250', '0.250']
attL_top25_sum : ['0.245', '0.268', '0.240', '0.248']
attL_top50_mean: ['0.250', '0.250', '0.250', '0.250']
attL_top50_sum : ['0.253', '0.266', '0.241', '0.241']
===== 4. Chosen action and reward
Action: talk to salesman
Reward: 0|Score: 5

——— ===== Step 11 ===== ———
===== 1. Textual obs:
o_desc: Souvenir Stand You find yourself standing in front of a
souvenir stand that is clearly affiliated with the Convention
Center. An odd−looking salesman sits behind the counter, his
head resting on his hand and his eyes glued to a spot
somewhere above and behind you. A notice on the counter points
toward a small, curtained room to the north. On the counter
are four candy bars (a Baby Rune, a Multi−Implementeers, a
ZM$100000 and a Forever Gores), some popcorn, some postcards
and a chicken finger. You can see a salesman here. The
salesman, losing interest in you, leans back and resumes
whatever it was he was doing (staring, mostly.)
o_inv: You are carrying: a cheaply−made sword a brass lantern (
providing light) The salesman, losing interest in you, leans
back and resumes whatever it was he was doing (staring, mostly
.)

```
o_feed: Although not much of a conversationalist, the salesman
    leans forward to listen. Two things are immediately obvious:
    1) He is not a regular user of toothpaste or mouthwash and 2)
    He is obviously wearing a mask.
a_past: talk to salesman
===== 2. newly extracted triplets
[('Souvenir Stand', 'has', 'exit to north'), ('all', 'in', '
    Souvenir Stand'), ('candy', 'in', 'Souvenir Stand'), ('counter
    ', 'in', 'Souvenir Stand'), ('finger', 'in', 'Souvenir Stand')
    , ('gores', 'in', 'Souvenir Stand'), ('he', 'is', 'regular'),
    ('he', 'is obviously wearing', 'mask'), ('he', 'is wearing', '
    mask'), ('his eyes', 'glued to', 'spot'), ('his eyes', 'glued
    to', 'spot above'), ('his eyes', 'glued to', 'spot somewhere
    above'), ('his head', 'resting on', 'his hand'), ('lantern', '
    in', 'Souvenir Stand'), ('looking salesman', 'sits behind', '
    counter'), ('mask', 'in', 'Souvenir Stand'), ('multi', 'in', '
    Souvenir Stand'), ('north', 'in', 'Souvenir Stand'), ('notice
    ', 'in', 'Souvenir Stand'), ('odd looking salesman', 'sits
    behind', 'counter'), ('odd salesman', 'sits behind', 'counter
    '), ('popcorn', 'in', 'Souvenir Stand'), ('postcards', 'in', '
    Souvenir Stand'), ('rune', 'in', 'Souvenir Stand'), ('salesman
    ', 'in', 'Souvenir Stand'), ('salesman', 'leans', 'listen'),
    ('salesman', 'leans forward', 'listen'), ('salesman', 'losing
    ', 'interest'), ('salesman', 'losing interest in', 'you'), ('
    salesman', 'sits behind', 'counter'), ('sword', 'in', '
    Souvenir Stand'), ('you', 'can see', 'salesman'), ('you', 'can
     see here', 'salesman'), ('you', 'have', ''), ('you', 'have',
    'brass lantern (providing light)'), ('you', 'have', 'cheaply−
    made sword'), ('you', 'have', 'salesman, losing interest in
    you, leans back and resumes whatever it was he was doing (
    staring, mostly.)')]
===== 3. Attention values:
————— attH: o_desc, o_inv, o_feed, a_past
attH_max        : ['0.252', '0.251', '0.248', '0.250']
attH_mean       : ['0.305', '0.231', '0.232', '0.232']
attH_sum        : ['1.000', '0.000', '0.000', '0.000']
attH_top10_mean: ['0.266', '0.242', '0.241', '0.251']
attH_top10_sum : ['0.429', '0.167', '0.160', '0.244']
attH_top25_mean: ['0.320', '0.225', '0.223', '0.232']
attH_top25_sum : ['0.999', '0.000', '0.000', '0.000']
attH_top50_mean: ['0.358', '0.213', '0.215', '0.215']
attH_top50_sum : ['1.000', '0.000', '0.000', '0.000']
————— attL: connectivity, item_in_room, item_in_inv, history
attL_max        : ['0.250', '0.250', '0.250', '0.250']
attL_mean       : ['0.250', '0.250', '0.250', '0.250']
attL_sum        : ['0.253', '0.251', '0.248', '0.248']
attL_top10_mean: ['0.250', '0.250', '0.250', '0.250']
attL_top10_sum : ['0.250', '0.252', '0.249', '0.249']
attL_top25_mean: ['0.250', '0.250', '0.250', '0.250']
attL_top25_sum : ['0.251', '0.253', '0.248', '0.248']
attL_top50_mean: ['0.250', '0.250', '0.250', '0.250']
attL_top50_sum : ['0.253', '0.251', '0.248', '0.248']
===== 4. Chosen action and reward
Action: throw mask
Reward: 8|Score: 13

————— ===== Step 12 ===== —————
===== 1. Textual obs:
```

o_desc: Souvenir Stand You find yourself standing in front of a souvenir stand that is clearly affiliated with the Convention Center. A notice on the counter points toward a small, curtained room to the north. On the counter are four candy bars (a Baby Rune, a Multi−Implementeers, a ZM$100000 and a Forever Gores), some popcorn, some postcards and a chicken finger.
o_inv: You are carrying: a cheaply−made sword a brass lantern (providing light)
o_feed: (first taking the mask) With the salesman's hot, fetid breath threatening to exfoliate multiple layers of facial epidermis, you quickly pull the mask from his face. In hellish harmony, two terrifying screams fill the air: yours, as the first human to see a grue; and his, as prelude to a graphic demonstration of why it is that they aren't seen by light of day (to wit: spontaneous grueish combustion, leaving only a greasy slick on the ground by your feet.) Although you will spend the rest of your life trying to forget what you have just seen, the sight of that sickly glowing fur, fish−mouthed face and razor−like fingers remains forever etched in your mind. You throw the mask away in horror at what you've just seen. [Your score has just gone up by eight points.]
a_past: throw mask
===== 2. newly extracted triplets
[('Souvenir Stand', 'has', 'exit to north'), ('Souvenir Stand', 'has', 'exit to up'), ('all', 'in', 'Souvenir Stand'), ('candy', 'in', 'Souvenir Stand'), ('counter', 'in', 'Souvenir Stand'), ('finger', 'in', 'Souvenir Stand'), ('fish', 'mouthed', 'face'), ('fish', 'mouthed', 'razor'), ('fish', 'mouthed face like', 'fingers'), ('fish', 'mouthed razor like', 'fingers'), ('forever', 'in', 'Souvenir Stand'), ('implementeers', 'in', 'Souvenir Stand'), ('lantern', 'in', 'Souvenir Stand'), ('north', 'in', 'Souvenir Stand'), ('notice', 'in', 'Souvenir Stand'), ('popcorn', 'in', 'Souvenir Stand'), ('postcards', 'in', 'Souvenir Stand'), ('rest', 'trying', 'forget'), ('rune', 'in', 'Souvenir Stand'), ('salesman', 'has', 'hot fetid breath'), ("salesman 's breath", 'exfoliate', 'layers'), ("salesman 's breath", 'exfoliate', 'layers of epidermis'), ("salesman 's breath", 'exfoliate', 'layers of facial epidermis'), ("salesman 's breath", 'exfoliate', 'multiple layers'), ("salesman 's breath", 'exfoliate', 'multiple layers of epidermis'), ("salesman 's breath", 'exfoliate', 'multiple layers of facial epidermis'), ("salesman 's fetid breath", 'exfoliate', 'layers'), ("salesman 's fetid breath", 'exfoliate', 'layers of epidermis'), ("salesman 's fetid breath", 'exfoliate', 'layers of facial epidermis'), ("salesman 's fetid breath", 'exfoliate', 'multiple layers'), ("salesman 's fetid breath", 'exfoliate', 'multiple layers of epidermis'), ("salesman 's fetid breath", 'exfoliate', 'multiple layers of facial epidermis'), ("salesman 's hot breath", 'exfoliate', 'layers'), ("salesman 's hot breath", 'exfoliate', 'layers of epidermis'), ("salesman 's hot breath", 'exfoliate', 'layers of facial epidermis'), ("salesman 's hot breath", 'exfoliate', 'multiple layers'), ("salesman 's hot breath", 'exfoliate', 'multiple layers of epidermis'), ("salesman 's hot breath", 'exfoliate', 'multiple layers of facial epidermis'), ("salesman 's hot fetid breath", 'exfoliate', 'layers'), ("salesman 's hot fetid breath", 'exfoliate', 'layers of epidermis'), ("salesman 's hot fetid breath", 'exfoliate', 'layers of

facial epidermis'), ("salesman 's hot fetid breath", '
exfoliate', 'multiple layers'), ("salesman 's hot fetid breath
", 'exfoliate', 'multiple layers of epidermis'), ("salesman 's
hot fetid breath", 'exfoliate', 'multiple layers of facial
epidermis'), ('sight', 'etched in', 'your mind'), ('sight', '
forever etched in', 'your mind'), ('sword', 'in', 'Souvenir
Stand'), ('terrifying screams', 'fill', 'air'), ('terrifying
screams', 'fill air in', 'harmony'), ('terrifying screams', '
fill air in', 'hellish harmony'), ('two terrifying screams', '
fill', 'air'), ('two terrifying screams', 'fill air in', '
harmony'), ('two terrifying screams', 'fill air in', 'hellish
harmony'), ('you', 'first taking', 'mask'), ('you', 'first
taking mask', "salesman 's breath threatening"), ('you', '
first taking mask', "salesman 's fetid breath threatening"),
('you', 'first taking mask', "salesman 's hot breath
threatening"), ('you', 'first taking mask', "salesman 's hot
fetid breath threatening"), ('you', 'have', 'brass lantern (
providing light)'), ('you', 'have', 'cheaply−made sword'), ('
you', 'pull', 'mask'), ('you', 'pull mask from', 'his face'),
('you', 'quickly pull', 'mask'), ('you', 'quickly pull mask
from', 'his face'), ('you', 'rest of', 'your life'), ('you', '
taking', 'mask'), ('you', 'taking mask', "salesman 's breath
threatening"), ('you', 'taking mask', "salesman 's fetid
breath threatening"), ('you', 'taking mask', "salesman 's hot
breath threatening"), ('you', 'taking mask', "salesman 's hot
fetid breath threatening"), ('you', 'throw', 'mask'), ('you',
'throw away', 'mask'), ('you', 'throw mask', "you 've just
seen"), ('you', 'throw mask', "you 've seen"), ('you', 'throw
mask away', "you 've just seen"), ('you', 'throw mask away', "
you 've seen"), ('you', 'throw mask away in', 'horror'), ('you
', 'throw mask in', 'horror'), ('you', 'will spend', 'rest'),
('you', 'will spend', 'rest of your life'), ('your score', '
has gone up by', 'eight points'), ('your score', 'has just
gone up by', 'eight points')]
===== 3. Attention values:
——— attH: o_desc, o_inv, o_feed, a_past
attH_max        : ['0.251', '0.251', '0.248', '0.250']
attH_mean       : ['0.298', '0.236', '0.232', '0.234']
attH_sum        : ['1.000', '0.000', '0.000', '0.000']
attH_top10_mean: ['0.260', '0.252', '0.239', '0.249']
attH_top10_sum : ['0.353', '0.265', '0.151', '0.231']
attH_top25_mean: ['0.311', '0.238', '0.221', '0.230']
attH_top25_sum : ['0.998', '0.001', '0.000', '0.001']
attH_top50_mean: ['0.348', '0.222', '0.214', '0.217']
attH_top50_sum : ['1.000', '0.000', '0.000', '0.000']
——— attL: connectivity, item_in_room, item_in_inv, history
attL_max        : ['0.249', '0.251', '0.250', '0.250']
attL_mean       : ['0.250', '0.250', '0.250', '0.250']
attL_sum        : ['0.250', '0.250', '0.250', '0.250']
attL_top10_mean: ['0.250', '0.250', '0.250', '0.250']
attL_top10_sum : ['0.246', '0.253', '0.249', '0.251']
attL_top25_mean: ['0.250', '0.250', '0.250', '0.250']
attL_top25_sum : ['0.245', '0.255', '0.248', '0.252']
attL_top50_mean: ['0.250', '0.250', '0.250', '0.250']
attL_top50_sum : ['0.250', '0.250', '0.250', '0.250']
===== 4. Chosen action and reward
Action: lower multi
Reward: 1|Score: 14

———— ===== Step 13 ===== ————
===== 1. Textual obs:
o_desc: Souvenir Stand You find yourself standing in front of a
    souvenir stand that is clearly affiliated with the Convention
    Center. A notice on the counter points toward a small,
    curtained room to the north. On the counter are three candy
    bars (a Baby Rune, a ZM$100000 and a Forever Gores), some
    popcorn, some postcards and a chicken finger. You can see a
    Multi−Implementeers here.
o_inv: You are carrying: a cheaply−made sword a brass lantern (
    providing light)
o_feed: (first taking the Multi−Implementeers) [Your score has
    just gone up by one point.] Dropped.
a_past: lower multi
===== 2. newly extracted triplets
[('Souvenir Stand', 'has', 'exit to north'), ('Souvenir Stand', '
    has', 'exit to up'), ('all', 'in', 'Souvenir Stand'), ('candy
    ', 'in', 'Souvenir Stand'), ('chicken', 'in', 'Souvenir Stand
    '), ('counter', 'in', 'Souvenir Stand'), ('gores', 'in', '
    Souvenir Stand'), ('lantern', 'in', 'Souvenir Stand'), ('multi
    ', 'in', 'Souvenir Stand'), ('north', 'in', 'Souvenir Stand'),
     ('notice', 'in', 'Souvenir Stand'), ('popcorn', 'in', '
    Souvenir Stand'), ('postcards', 'in', 'Souvenir Stand'), ('
    rune', 'in', 'Souvenir Stand'), ('sword', 'in', 'Souvenir
    Stand'), ('you', 'can see', 'multi implementeers'), ('you', '
    can see here', 'multi implementeers'), ('you', 'have', 'brass
    lantern (providing light)'), ('you', 'have', 'cheaply−made
    sword'), ('your score', 'has gone up by', 'one point'), ('your
     score', 'has just gone up by', 'one point')]
===== 3. Attention values:
———— attH: o_desc, o_inv, o_feed, a_past
attH_max        : ['0.250', '0.250', '0.250', '0.250']
attH_mean       : ['0.299', '0.236', '0.235', '0.230']
attH_sum        : ['1.000', '0.000', '0.000', '0.000']
attH_top10_mean: ['0.258', '0.252', '0.243', '0.247']
attH_top10_sum : ['0.330', '0.268', '0.185', '0.217']
attH_top25_mean: ['0.312', '0.240', '0.228', '0.221']
attH_top25_sum : ['0.998', '0.001', '0.000', '0.000']
attH_top50_mean: ['0.351', '0.220', '0.219', '0.210']
attH_top50_sum : ['1.000', '0.000', '0.000', '0.000']
———— attL: connectivity, item_in_room, item_in_inv, history
attL_max        : ['0.248', '0.252', '0.249', '0.252']
attL_mean       : ['0.250', '0.249', '0.250', '0.251']
attL_sum        : ['0.242', '0.219', '0.245', '0.294']
attL_top10_mean: ['0.248', '0.251', '0.249', '0.252']
attL_top10_sum : ['0.234', '0.257', '0.240', '0.269']
attL_top25_mean: ['0.249', '0.250', '0.249', '0.251']
attL_top25_sum : ['0.225', '0.253', '0.234', '0.288']
attL_top50_mean: ['0.250', '0.249', '0.250', '0.251']
attL_top50_sum : ['0.242', '0.219', '0.245', '0.294']
===== 4. Chosen action and reward
Action: lower multi
Reward: 1|Score: 15

———— ===== Step 14 ===== ————
===== 1. Textual obs:
o_desc: Souvenir Stand You find yourself standing in front of a
    souvenir stand that is clearly affiliated with the Convention
    Center. A notice on the counter points toward a small,

curtained room to the north. On the counter are three candy
    bars (a Baby Rune, a ZM$100000 and a Forever Gores), some
    popcorn, some postcards and a chicken finger. You can see a
    Multi−Implementeers here.
o_inv: You are carrying: a cheaply−made sword a brass lantern (
    providing light)
o_feed: (first taking the Multi−Implementeers) [Your score has
    just gone up by one point.] Dropped.
a_past: lower multi
===== 2. newly extracted triplets
[('Souvenir Stand', 'has', 'exit to north'), ('Souvenir Stand', '
    has', 'exit to up'), ('all', 'in', 'Souvenir Stand'), ('candy
    ', 'in', 'Souvenir Stand'), ('chicken', 'in', 'Souvenir Stand
    '), ('counter', 'in', 'Souvenir Stand'), ('gores', 'in', '
    Souvenir Stand'), ('lantern', 'in', 'Souvenir Stand'), ('multi
    ', 'in', 'Souvenir Stand'), ('north', 'in', 'Souvenir Stand'),
     ('notice', 'in', 'Souvenir Stand'), ('popcorn', 'in', '
    Souvenir Stand'), ('postcards', 'in', 'Souvenir Stand'), ('
    rune', 'in', 'Souvenir Stand'), ('sword', 'in', 'Souvenir
    Stand'), ('you', 'can see', 'multi implementeers'), ('you', '
    can see here', 'multi implementeers'), ('you', 'have', 'brass
    lantern (providing light)'), ('you', 'have', 'cheaply−made
    sword'), ('your score', 'has gone up by', 'one point'), ('your
     score', 'has just gone up by', 'one point')]
===== 3. Attention values:
——— attH: o_desc, o_inv, o_feed, a_past
attH_max        : ['0.250', '0.250', '0.250', '0.250']
attH_mean       : ['0.299', '0.238', '0.232', '0.231']
attH_sum        : ['1.000', '0.000', '0.000', '0.000']
attH_top10_mean: ['0.258', '0.252', '0.243', '0.247']
attH_top10_sum : ['0.334', '0.267', '0.180', '0.218']
attH_top25_mean: ['0.312', '0.245', '0.218', '0.225']
attH_top25_sum : ['0.997', '0.002', '0.000', '0.000']
attH_top50_mean: ['0.351', '0.224', '0.213', '0.211']
attH_top50_sum : ['1.000', '0.000', '0.000', '0.000']
——— attL: connectivity, item_in_room, item_in_inv, history
attL_max        : ['0.249', '0.251', '0.249', '0.252']
attL_mean       : ['0.250', '0.249', '0.250', '0.251']
attL_sum        : ['0.249', '0.224', '0.246', '0.281']
attL_top10_mean: ['0.249', '0.251', '0.249', '0.251']
attL_top10_sum : ['0.239', '0.256', '0.242', '0.263']
attL_top25_mean: ['0.249', '0.250', '0.250', '0.251']
attL_top25_sum : ['0.235', '0.250', '0.238', '0.277']
attL_top50_mean: ['0.250', '0.249', '0.250', '0.251']
attL_top50_sum : ['0.249', '0.224', '0.246', '0.281']
===== 4. Chosen action and reward
Action: lower gores
Reward: 1|Score: 16

——— ===== Step 15 ===== ———
===== 1. Textual obs:
o_desc: Souvenir Stand You find yourself standing in front of a
    souvenir stand that is clearly affiliated with the Convention
    Center. A notice on the counter points toward a small,
    curtained room to the north. On the counter are two candy bars
     (a Baby Rune and a ZM$100000), some popcorn, some postcards
    and a chicken finger. You can see two candy bars (a Forever
    Gores and a Multi−Implementeers) here.

```
o_inv: You are carrying: a cheaply-made sword a brass lantern (
    providing light)
o_feed: (first taking the Forever Gores) [Your score has just gone
    up by one point.] Dropped.
a_past: lower gores
===== 2. newly extracted triplets
[('Souvenir Stand', 'has', 'exit to north'), ('Souvenir Stand', '
    has', 'exit to up'), ('all', 'in', 'Souvenir Stand'), ('candy
    ', 'in', 'Souvenir Stand'), ('chicken', 'in', 'Souvenir Stand
    '), ('counter', 'in', 'Souvenir Stand'), ('gores', 'in', '
    Souvenir Stand'), ('lantern', 'in', 'Souvenir Stand'), ('multi
    ', 'in', 'Souvenir Stand'), ('north', 'in', 'Souvenir Stand'),
     ('notice', 'in', 'Souvenir Stand'), ('popcorn', 'in', '
    Souvenir Stand'), ('postcards', 'in', 'Souvenir Stand'), ('
    rune', 'in', 'Souvenir Stand'), ('sword', 'in', 'Souvenir
    Stand'), ('you', 'can see', 'two candy bars'), ('you', 'can
    see here', 'two candy bars'), ('you', 'have', 'brass lantern (
    providing light)'), ('you', 'have', 'cheaply-made sword'), ('
    your score', 'has gone up by', 'one point'), ('your score', '
    has just gone up by', 'one point')]
===== 3. Attention values:
------ attH: o_desc, o_inv, o_feed, a_past
attH_max        : ['0.250', '0.250', '0.250', '0.250']
attH_mean       : ['0.294', '0.234', '0.252', '0.219']
attH_sum        : ['1.000', '0.000', '0.000', '0.000']
attH_top10_mean: ['0.261', '0.254', '0.254', '0.231']
attH_top10_sum : ['0.353', '0.271', '0.272', '0.104']
attH_top25_mean: ['0.313', '0.230', '0.273', '0.184']
attH_top25_sum : ['0.966', '0.000', '0.034', '0.000']
attH_top50_mean: ['0.342', '0.217', '0.251', '0.189']
attH_top50_sum : ['1.000', '0.000', '0.000', '0.000']
------ attL: connectivity, item_in_room, item_in_inv, history
attL_max        : ['0.248', '0.251', '0.249', '0.251']
attL_mean       : ['0.250', '0.250', '0.250', '0.250']
attL_sum        : ['0.248', '0.229', '0.248', '0.275']
attL_top10_mean: ['0.249', '0.251', '0.250', '0.251']
attL_top10_sum : ['0.239', '0.256', '0.246', '0.260']
attL_top25_mean: ['0.249', '0.250', '0.250', '0.251']
attL_top25_sum : ['0.233', '0.255', '0.242', '0.270']
attL_top50_mean: ['0.250', '0.250', '0.250', '0.250']
attL_top50_sum : ['0.248', '0.229', '0.248', '0.275']
===== 4. Chosen action and reward
Action: put multi on gores
Reward: 0|Score: 16
```