[Reviews · NeurIPS 2020]

Review 1

Summary and Contributions: This paper focused on learning a policy for text-based games. The main contributions are two parts: 1) proposed to use knowledge graph to record history, and proposed to divide knowledge graph to sub-graphs to model relational and temporal awareness; 2) proposed hierarchical attention networks, in which the high level attention uses the full knowledge graph and the low level attention uses sub-graphs.

Strengths: POST REBUTTAL: I agree with the other reviewer that the paper has clear empirical gains, but the methodological contribution is not fundamental. Therefore I remain my score of 6. 1. The empirical results show that the proposed sub-graph division and hierarchical attention work well and achieve state-of-the-art performance. 2. The idea of sub-graph division is interesting.

Weaknesses: 1. The paper relies on pre-defined rules (Section 5.2) to divide the full knowledge graph to sub-graphs, which make the paper less interesting. 2. Although the paper has an ablation study, it does not contain enough empirical study and discussion about sub-graphs. For example, in Section 4.2, the authors mentioned that there are two type of sub-graphs, one with historical information and one does not. Which one contributes the most to the final performance? There are two awareness mentioned in Section 4.2 (temporal-awareness and relational awareness), which one contributes the most to the final performance, and why? What exactly is missing from full knowledge graph that sub-graph captures? There should be more discussion about why sub-graphs and low-level attention helps.

Correctness: yes

Clarity: yes

Relation to Prior Work: yes

Reproducibility: Yes

Additional Feedback:


Review 2

Summary and Contributions: This paper studies the creation of RL agents for text-based games and specifically proposes a stacked hierarchical attention-based knowledge graph to allow the agent to reason about these games. In contrast to previous work in this domain that used knowledge graphs, the proposed approach has better performance due to the mutli-level and multi-modal reasoning. This approach is validated on a variety of man-made text games and shows promising improvements over existing agents.

Strengths: This work extends the KG-A2C agent in several ways: First it uses attention to re-weight the different components of textual observations (e.g. narrative, inventory, location description, and previous action text). The output of this first attention is then combined with another attention computed over multiple different sub-knowledge graphs corresponding to the connectivity of locations, objects in the current location, inventory, and anything that is connected to the current player. While none of the individual building blocks are particularly novel, the combination of all of these elements introduces a lot of flexibility to structurally decompose the different types of knowledge available in the game and allow the agent to pay attention to specific subsets of this knowledge. This flexibility pays dividends when it comes to the experimental evaluation and where this agent significantly improves on KG-A2C in nearly every game. The ablations presented validate that the full stacked architecture is indeed needed to maintain current levels of performance, and the analysis shows that the attention mechansims are working well insofar as they distribute attention correctly between locations descriptions and inventory contents as needed to generate the action. Overall I'm most impressed by the nearly universal improvements achieved by this method.

Weaknesses: While the improvements in game score are undeniable, I find it harder to understand why the SHA-KG architecture is leading to higher scores. Possible hypotheses are the multiple sub-KGs, the attention based processing of observations, the two levels of hierarchy, or some combination herein. While ablations give the idea that the overall architecture is necessary, it's hard to distill which parts (e.g. relational awareness vs historical awareness) are actually leading to better performance and how future agents can build or extend this architecture.

Correctness: As far as I can tell.

Clarity: The paper suffers from minor grammar mistakes, which could hopefully be addressed given further time.

Relation to Prior Work: Yes

Reproducibility: Yes

Additional Feedback: No source code is provided - so it may be difficult to exactly reproduce the agent. However the detailed equations should be a pretty good guide. After reading the rebuttal and in light of the reviewer discussion - I still think this paper is above acceptance threshold and maintain my original scoring.


Review 3

Summary and Contributions: The paper presents an approach to playing text-based games using knowledge graph. Previous works show how to use KGs to deal with partial observability, large action spaces, etc. and this work focuses on giving the agent the ability to reason over the KG and input text descriptions with a stacked hierarchical attention mechanism.

Strengths: 1. The paper presents a contribution that directly fixes an inherent limitation in the work that it is based on (the KG-A2C) and other works using KGs for text games – the agents are not using the graph to its fullest potential. 2. The inspiration to use techniques from VQA and treat the KG as another modality is well explained and validated. 3. Results show good improvement over the current state of the art across a suite of games. An ablation study validates most of the architecture choices and provides evidence for both layers of the hierarchy.

Weaknesses: 1. There’s some controversy over whether attention values actually mean anything when applied to text and similar arguments apply to knowledge graphs as well (see “Attention is Not Explanation” https://arxiv.org/abs/1902.10186 NAACL-19 and “Attention is Not Not Explanation” https://arxiv.org/abs/1908.04626 EMNLP-19). Given this and the author’s claims regarding SHA-KG “reasoning” and being interpretable, I do not think that current evidence sufficiently backs the former claim at least. 1a. I would further suggest a reframing of the writing to reflect that this is faithful interprebility at most, i.e. pointing out likely places in text/KG where the action is made (see this for more details https://dl.acm.org/doi/10.1145/3236009) 2. What exactly is the reasoning happening on the KGs here? Some of the confusion is in part because I do not understand what each of the sub-graphs mean semantically. I see examples in the supplementary but the inter play between the sub-graphs and why it was partitioned in this way is lost on me. Some examples of perhaps which nodes on the graph "light up" when making a certain decision would help here and/or ablations differences in attention with varying numbers of sub-graphs might help here

Correctness: A key component of the original KG-A2C is that the knowledge graph was used to constrain the action space via a “graph mask” – I do not see any mentions of whether this mask was used. Although the mask was effective, some games had better performance without it. If it was used, details such as dropout on the mask, etc. should be given in addition to an ablation of SHA-KG without using the mask. And vice versa, if it wasn’t used, then how does SHA-KG do with it. Without this ablation, it is hard to tell exactly how much of the gain in performance can be attributed to the state representation architecture.

Clarity: The paper is well written overall, but I think some changes can be made to improve understanding. 1. Move some details regarding semantics of the graph partitions to the main paper with an example or two, perhaps at the expense of hyperparameter details. 2. Reformulate some of the claims regarding reasoning/interprebility as I have written before. Minor: line 107 rephrase: “In a …, it requires” -> “Text-based games require”

Relation to Prior Work: The main difference between the base work KG-A2C and this is the architecture that encodes the knowledge graph. This is clearly stated and choices for this architecture are explained.

Reproducibility: Yes

Additional Feedback: The paper contributes, in my opinion, one main useful thing - it shows how to adapt architectures from neighboring fields such as vision&language to treat KGs as an extra modality in traditionally text-only settings. It is a simple but effective idea with clear empirical gains and fills in a short coming of KG-A2C. The main issue I have with this work is with respect to their claims of interpretability, sure you can highlight a few attention values from the GATs but you cannot make any claims regarding "reasoning" just from that as they seem to do. They also do not position their paper and terminology used with respect to all the other work in terms of interpretability/explainability. I put in a few pointers in my original review and the authors said they'd cite it but without seeing what the rewrite of that section looks like, its hard to tell. For these reasons, I will maintain my score of 6 after reading the rebuttal.


Review 4

Summary and Contributions: This paper describes an approach for solving text-based games. Its main contribution is to use knowledge graph reasoning to address this task. The paper describes a two-step attention mechanism that reasons over subgraphs of the knowledge graph independently. The system and baselines are evaluated on several challenging text-based games.

Strengths: The proposed method is described clearly and shows obvious improvements over existing baselines. The paper also describes some interpretability analysis of the proposed method on several games using the attention mechanism.

Weaknesses: The main concern I have with the proposed approach is its extensibility. How could the use of a knowledge graph extend to more continuous and realistic environments, such as games that include visual observations? And how much of a problem was error propagation in the evaluation of this system (e.g., errors made by OpenIE -- it would be good to have some quantitative analysis, like sampling of extracted triples, to get a feel of how well OpenIE works on these games)? After the rebuttal and discussion, I still think positively of the paper. However, I am also taking into account and agree with the points other reviewers brought up about discussing more the interpretability and reasoning aspects.

Correctness: The system is evaluated on a difficult set of text-based games. There were a few questions I had about the evaluation: * What does "maximum score" mean? Is that like the maximum possible score in the game? * Are different systems trained separately on the 20 games, or is a single system expected to multi-task across games? Regardless, it would be interesting to see a multitasking setup where the system is trained on one subset of the games and tested on another set.

Clarity: The formal definition of the model is very clear. A few minor suggestions: * In the abstract, "explicit reasoning with *a* knowledge graph" sounds a bit better to me because it's not clear what "the knowledge graph" refers to until I've read the intro. * I didn't understand the point made on line 40-41 (last sentence) * In line 135, shouldn't s_t be o_t (observation)? Maybe I'm misunderstanding but s is the true state, right? So the system should only be encoding the observation * What is a "component" of a textual observation? (L150) This became clearer later in the paper, but not super clear without closely knowing the evaluation data. * I'm not very familiar with text-based games, but are the games referenced in [13] really not made by humans? Or is it that Jericho includes games that are intended for humans to play, rather than games designed for evaluating RL agents? If so, probably "man-made" is a bit misleading and something like "developed for humans to play" would be more accurate. * Would be good to give some context about the three example games in Figure 3 / Section 5.5 -- like what are the general themes/contexts of the games?

Relation to Prior Work: I am not aware of any work that is missing, although I am not very familiar with environments/systems specifically for text-based games.

Reproducibility: Yes

Additional Feedback: * Why did you reduce the node embedding dimension from KG-A2C?

[Author Response · NeurIPS 2020]

We thank all reviewers for their valuable comments and suggestions. We'll incorporate suggestions and clarifications in the revision. We first address a shared point (by Reviewer 1 and 2) and then respond to each reviewer respectively.

**Further ablation about sub-graphs (R1 and R2).** We have provided an ablation study about the main components of our method (e.g., multi-level attention and group attention) in the main paper. Regarding the detailed contributions of different types of sub-graphs, we further design three variants with different graph partitioning strategies: 1) "w/o relation awareness" combines $o_{KG,2}$ (room objects) and $o_{KG,3}$ (collected objects). 2) "w/o history awareness" combines $o_{KG,4}$ with $o_{KG,2}$ and $o_{KG,3}$, respectively. 3) "w/o history" removes all historical information. Results in following figures indicate that the effect of different types of awareness varies with respect to the games. No simple conclusions can be made regarding which type of awareness contributes the most to the final performance (e.g., "w/o relation awareness" and "w/o history awareness" behave differently in "zork1" and "zork3"). However, considering them collectively and learning to balance their importance lead to the improved performance of our method.

Reviewer 1 **Q1: Pre-defined rules. A1:** Our method does not have to rely on pre-defined rules to partition a knowledge graph. In fact, subgraphs are allowed to be constructed using different approaches. As there is no previous work considering subgraphs in text-based games, for simplicity, we use rules for graph partitioning, which can explicitly distinguish information and provide interpretability for the behaviors of the agent. In the future, we are interested in investigating other partitioning methods, such as language model-based question-answering (although the questions should be predefined), and automated partitioning.
**Q2: What is missing from full KG that sub-graph captures? A2:** The full KG can not distinguish between the current information and the historical information (note that SHA-KG also uses full KG in the high level). An example has been provided in Supplementary's Fig. 2 (b), where the yellow part denotes the current information and the green part indicates the historical information. From the yellow part, we can not tell whether an object has been collected (in inventory) or not. As shown in Fig. 2 (c), applying sub-graphs enables us to explicitly capture such information.

Reviewer 2 **Q1: Why SHA-KG architecture is leading to higher scores? A1:** Our two-level attention mechanism provides an explainable way to refine information. The first level of hierarchy tells the agent which part of the textual information should be focused on. Based on the output of this level, the second level of hierarchy informs the agent which part of a knowledge graph should be targeting.
**Q2: Minor grammar mistakes and code. A2:** We will address the grammar mistakes in the final version. Our code will be made publicly available upon publication.

Reviewer 3 **Q1: Controversy on attention. A1:** We will add discussion in the revision and cite the relevant papers.
**Q2: What is the reasoning happening on KGs? A2:** We will add an example to illustrate the graph partition and attention assignment, as suggested by the reviewer. Each graph indicates different types of information. In the right figure (based on Fig. 3 ztuu), for example, $o_{KG,1}$ indicates room connectivity. We omit $o_{KG,4}$ due to space limit. Our SHA helps the agent to use information efficiently for taking actions. The digit under the

sub-graph denotes graph-level attention. Since the $o_{KG,2}$ has the highest attention, the agent will focus more on objects it contains. In each sub-graph, nodes with top-3 highest attention (by GATs) are highlighted in yellow. Such node-level attention helps to further constrain (softly) the objects in $o_{KG,2}$ to derive the actions.
**Q3: Graph mask for action selection. A3:** All KG-related models construct masks from $o_{KG, full}$. As the benefit of graph mask has been investigated in KG-A2C, we use the same action selection strategy to make the comparison fair.

Reviewer 4 **Q1: Extensibility. A1:** Our method can be extended to handle continuous and realistic environments, if a knowledge graph can be constructed. For example, the entities of a knowledge graph may be extracted via object detection techniques for a visual environment. However, our hierarchical attention module is general to most tasks.
**Q2: OpenIE. A2:** We will add quantitative analysis for information extraction error.
**Q3: Meaning of maximum score. A3:** Yes, it refers to the maximum possible score in the game.
**Q4: Multi-task setting. A4:** The games used in this work have quite different characteristics, making it hard to share knowledge. Compared with Jericho, TextWorld may be a more suitable testbed for studying MTL, since it enables to generate a set of similar games. We leave the MTL as a future work.
**Q5: Minor suggestions. A5:** We will refine our paper based on these suggestions.

[Meta-Review · NeurIPS 2020]

The work is an interesting approach of extending KG-A2C with sub-graphs to achieve impressive state of the art performance on several games. Ablation studies show that this architecture is needed to achieve the performance and the attention analysis is interesting. The work could benefit from a more thorough analysis of what the model is doing (beyond just attention values which are questionable). The paper could also benefit from improved clarity in its writing.